# KNOWLEDGE-AUGMENTED LONG-CoT GENERATION FOR COMPLEX BIOMOLECULAR REASONING

## ABSTRACT

Understanding complex biomolecular mechanisms requires multi-step reasoning across molecular interactions, signaling cascades, and metabolic pathways. While large language models (LLMs) show promise in such tasks, their application to biomolecular problems is hindered by logical inconsistencies and the lack of grounding in domain knowledge. Existing approaches often exacerbate these issues: reasoning steps may deviate from biological facts or fail to capture long mechanistic dependencies. To address these challenges, we propose a Knowledge-Augmented Long-CoT Reasoning framework that integrates LLMs with knowledge graph–based multi-hop reasoning chains. The framework constructs mechanistic chains via guided multi-hop traversal and pruning on the knowledge graph; these chains are then incorporated into supervised fine-tuning to improve factual grounding and further refined with reinforcement learning to enhance reasoning reliability and consistency. Furthermore, to overcome the shortcomings of existing benchmarks, which are often restricted in scale and scope and lack annotations for deep reasoning chains, we introduce PrimeKGQA, a comprehensive benchmark for biomolecular question answering. Experimental results on both PrimeKGQA and existing datasets demonstrate that although larger closed-source models still perform well on relatively simple tasks, our method demonstrates clear advantages as reasoning depth increases, achieving state-of-the-art performance on multi-hop tasks that demand traversal of structured biological knowledge. These findings highlight the effectiveness of combining structured knowledge with advanced reasoning strategies for reliable and interpretable biomolecular reasoning.

## 1 INTRODUCTION

Biological systems are governed by extraordinarily complex mechanisms spanning multiple levels, from protein–protein interactions to intracellular signaling cascades and metabolic pathways (Stoney et al., 2018). Understanding these mechanisms is crucial for areas such as drug discovery, elucidation of disease mechanisms, and analysis of molecular interactions. Crucially, molecular events rarely occur in isolation; instead, they operate within intricate causal chains, making multi-step reasoning indispensable for interpretable mechanistic insights and scientific discovery (Patel et al., 2005; Xu et al., 2024). Meanwhile, the rapid expansion of genomic and proteomic datasets (Ma et al., 2023; Hosseini et al., 2024; Fallahpour et al., 2025), poses unprecedented challenges for computational methods to extract reliable and interpretable reasoning from large-scale biological data.

Recent advances in Large Language Models (LLMs) (Hurst et al., 2024; Guo et al., 2025; Yang et al., 2025) have recently achieved remarkable progress in multi-step reasoning tasks, particularly in domains such as mathematics (Yu et al., 2023), logic (Chen et al., 2024a), and programming (El-Kishky et al., 2025). These advances are largely attributed to Chain-of-Thought (CoT) (Wei et al., 2022; Wang et al., 2022; Kojima et al., 2022; Min et al., 2024) prompting techniques and reinforcement learning strategies, which enable stepwise and logically coherent reasoning. These advances highlight the potential of LLMs to support mechanistic reasoning in biomolecular research, where causal dependencies and multi-hop knowledge traversal are central.

However, direct application of LLMs to biomolecular reasoning remains highly non-trivial (Li et al., 2024; Zhuang et al., 2025): models can produce biologically implausible outputs (Zheng et al., 2024; Fang et al., 2023), generate reasoning chains with logical inconsistencies (Zhang et al., 2024), and

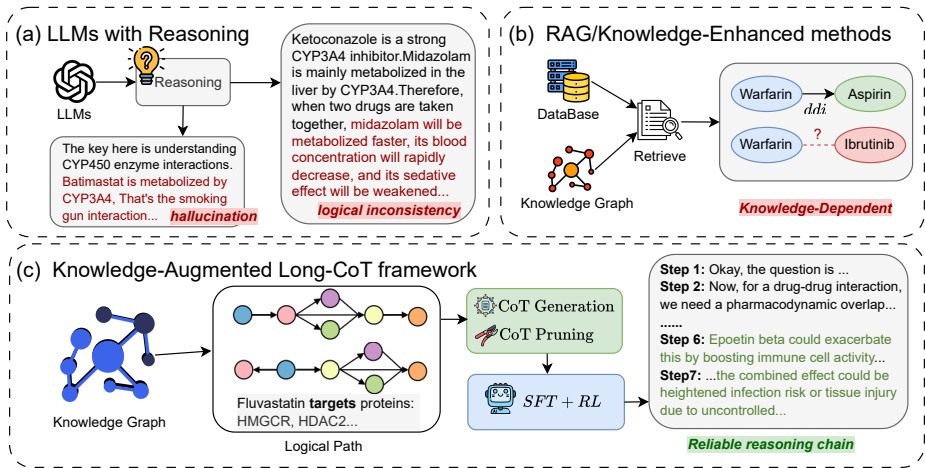

Figure 1: Comparison of different reasoning approaches for biomolecular problems. (a) Reasoning LLMs generate multi-step reasoning chains but often suffer from hallucinations and logical inconsistencies. (b) Retrieval-augmented generation and related knowledge-enhanced methods reduce hallucinations but are knowledge-dependent on the quality and coverage of sources. (c) Our proposed Knowledge-Augmented Long-CoT framework integrates knowledge graph-guided reasoning, enabling logically coherent and reliable reasoning chains for complex biomolecular tasks.

fail to leverage structured knowledge inherent in biological systems (Figure 1a). Although recent knowledge-guided generation methods, such as retrieval-augmented (Soman et al., 2024; Li et al., 2025) or knowledge-graph-based approaches (Liu et al., 2021; Luo et al., 2024; Zhao et al., 2025), partially alleviate hallucination and factual errors, but are inherently knowledge-dependent, relying on the coverage and reliability of external sources (Figure 1b). **Addressing these limitations requires a framework that prioritizes the principled utilization of existing structured knowledge to guide model reasoning, rather than the pursuit of knowledge completeness, thereby enhancing LLM inference and enabling reliable, domain-grounded conclusions.**

To address these challenges, we propose Bio-KCoT, a knowledge-augmented long-CoT reasoning framework for complex biomolecular problems (Figure 1c). At its core, Bio-KCoT introduces a knowledge graph–guided reasoning method that integrates structured biological knowledge to generate biologically plausible and logically coherent reasoning paths. Because raw knowledge graphs contain many redundant or spurious connections, we employ systematic path search and pruning to retain only reasoning chains that are both biologically meaningful and logically complete. These curated chains are then incorporated into supervised fine-tuning to strengthen factual grounding and reinforcement learning with Group Relative Policy Optimization (GRPO) to refine reasoning strategies and enhance robustness on challenging multi-hop tasks. Finally, to enable systematic evaluation, we introduce PrimeKGQA, a benchmark derived from the PrimeKG knowledge graph (Chandak et al., 2023), which spans diverse biomolecular question answering scenarios and task types. PrimeKGQA provides a standardized basis for fair comparison against strong baselines and supports progress in biologically grounded multi-step reasoning. We summarize our contributions as follows:

- We introduce Bio-KCoT, a knowledge-augmented long-CoT reasoning framework for biomolecular problems. It integrates KG-guided path search and pruning to generate coherent reasoning chains used during supervised fine-tuning and reinforcement learning.

- We introduce a high-quality benchmark for biomolecular reasoning, built by systematically collecting and curating knowledge graph-guided reasoning paths that capture multi-level semantic relationships. To ensure fair evaluation, the dataset encompasses diverse biomolecular reasoning scenarios and task types.

- Experimental results on both PrimeKGQA and existing datasets show that while larger closed-source models may retain advantages on simpler tasks, our method achieves substantial gains as reasoning complexity increases, delivering the strongest performance on multi-hop tasks requiring traversal of structured biological knowledge.

## 2 RELATED WORK

In this section, we provide a comprehensive review of related work, highlighting prior approaches and their relevance to biomolecular reasoning.

### 2.1 CHAIN-OF-THOUGHT REASONING

Chain-of-Thought reasoning enables LLMs to generate intermediate reasoning steps before arriving at a final answer (Wei et al., 2022; Kojima et al., 2022). This paradigm not only improves the transparency of model predictions, but also enhances performance on tasks requiring multi-step reasoning. Early work introduced few-shot CoT prompting with manually crafted demonstrations (Wei et al., 2022), or simple zero-shot prompting such as "Let's think step by step" (Kojima et al., 2022). To further strengthen the quality of reasoning, several extensions have been proposed, including self-consistency (Wang et al., 2022), least-to-most prompting (Zhou et al., 2022), complexity-based prompting (Fu et al., 2022), and self-polishing techniques (Xi et al., 2023). More recently, CoT reasoning has been explicitly incorporated into the design of advanced reasoning-oriented LLMs, leading to the emergence of models such as o1 (OpenAI, 2024), DeepSeek-R1 (Guo et al., 2025), and Google's Gemini 2.5 Pro (Comanici et al., 2025), which integrate dedicated "thinking" mechanisms and training strategies to enhance multi-step reasoning, coding, and scientific problem solving. Concurrently, the emergence of reasoning-oriented LLMs stimulated interest in distilling their capabilities into smaller student models (<10B), where teacher-generated CoTs are employed as effective supervision signals (Ho et al., 2022; Fu et al., 2023; Magister et al., 2022). However, distilled models still struggle to generalize reasoning beyond benchmark datasets, particularly when factual grounding is critical.

### 2.2 SCIENTIFIC AND BIOMOLECULAR CoT REASONING

Beyond general NLP benchmarks, the CoT paradigm is increasingly being adapted for complex reasoning in scientific domains. In medicine, models like HuatuoGPT (Chen et al., 2024a) and ReasonMed (Sun et al., 2025) leverage multi-stage, knowledge-infused pipelines to enhance diagnostic reliability (Muennighoff et al., 2025; Dutta & Hsiao, 2024; Zuo et al., 2025). This trend is mirrored in the biomolecular sciences, where models are being developed to reason about intricate biological and chemical processes. These efforts range from incorporating fundamental molecular features like DNA sequences (Fallahpour et al., 2025), protein evolutionary profiles (Liu et al., 2025), and molecular structures (M. Bran et al., 2024; Jang et al., 2024; Narayanan et al., 2025), to modeling higher-level systems such as protein-protein interaction pathways (Jin et al., 2024), tree-structured biological processes (Hsu et al., 2024), and single-cell annotations (Fang et al., 2025).

To facilitate progress, the community relies on benchmarks such as ScienceQA (Lu et al., 2022), MedQA (Jin et al., 2021), PubMedQA (Jin et al., 2019), and BioASQ (Krithara et al., 2023). More recently, pathway-centric datasets have been introduced to specifically challenge reasoning over biochemical interactions (Li et al., 2023; Park et al., 2025; Zhao et al., 2025). However, these existing resources often lack the explicit, multi-step rationales tailored to biomolecular reasoning. This scarcity of detailed ground-truth reasoning chains limits their effectiveness in training the very long-CoT models needed to tackle deep biological questions.

### 2.3 KNOWLEDGE-AUGMENTED REASONING

Although CoT provides a reasoning scaffold, its reliability is fundamentally limited by the internal knowledge of the LLM, which can be prone to hallucination (Huang et al., 2025; Griot et al., 2025; Chen et al., 2024b). To address this limitation, recent work has explored knowledge augmentation, which grounds reasoning in external verified sources such as domain-specific databases or KGs (Chen et al., 2024a; Xie et al., 2024). Among these, KGs are particularly powerful as they provide structured knowledge about entities and their relationships, which can be integrated into LLM reasoning through various techniques, from embedding-based methods to modern hybrid frameworks that enable direct interaction between the LLM and the graph (Guo et al., 2024; Wang et al., 2023; Chandak et al., 2023). This ability to represent complex relational structures makes KGs particularly valuable in domains such as biomedicine, which are defined by intricate networks of interactions.

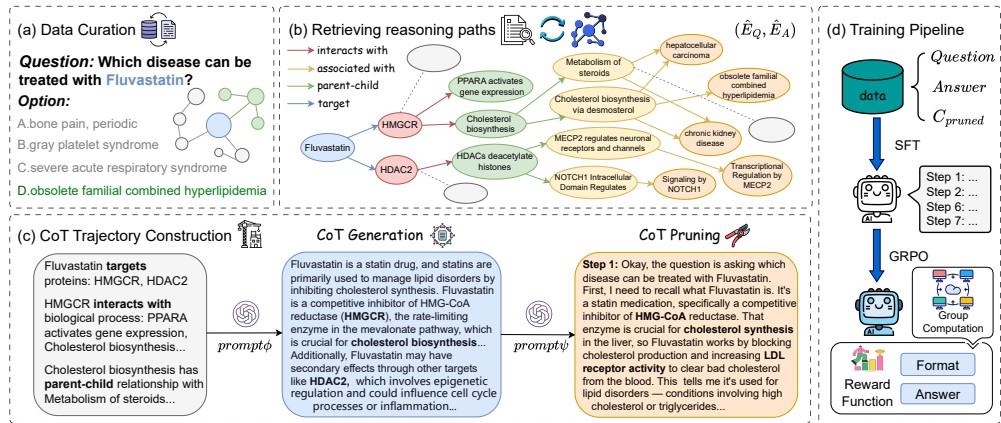

Figure 2: Overview of our proposed framework. (a) **Data Curation**: Given a biomolecular question and candidate answers, we extract entities from the question ($E_Q$) and the correct answer ($E_A$). (b) **Retrieving Reasoning Paths**: The extracted entities are mapped onto KG nodes. Reasoning paths $\mathcal{P}(Q, A; d)$ are then retrieved using predefined templates. (c) **CoT Trajectory Construction**: The induced path $p$ provides semantic relations that guide the initial CoT generation. The generated trajectories are further refined through the pruning stage to ensure clarity and accuracy. (d) **Training Pipeline**: The curated $(Q, A, C_{\text{pruned}})$ pairs are used for supervised fine-tuning (SFT), followed by reinforcement learning (GRPO) to align the model's reasoning and answer generation.

For biomolecular reasoning, where accurate modeling of interactions among genes, proteins, and pathways is essential, knowledge-augmented approaches are particularly promising (Liu et al., 2021; Luo et al., 2024; Zhao et al., 2025). While approaches such as MedReason (Wu et al., 2025) and KG-o1 (Wang et al., 2025) leverage KGs for supervised fine-tuning through shortest-path–based extraction, this strategy is generally well-suited for tasks where relevant evidence can indeed be captured within short paths. However, its effectiveness diminishes for complex problems that require integrating knowledge from disparate parts of the graph. By constraining the evidence to a local neighborhood, it not only risks oversimplifying the reasoning process but also fails to support robust, multi-step inference.

## 3 METHODOLOGY

The overall workflow, including data curation, reasoning path retrieval, CoT generation, and training pipeline, is illustrated in Figure 2.

### 3.1 RETRIEVING REASONING PATHS

Given a biomolecular question $Q$ (with problem statement and candidate options) and its corresponding correct answer $A$, our first step is to retrieve reasoning paths from the pre-constructed biomolecular KG $G = (V, E)$, where $V$ denotes the set of entities and $E$ the relations. The goal is to extract structured reasoning chains $\mathcal{P}$ that bridge the question and the answer.

**Entity extraction and mapping.** We extract entities mentioned in $Q$ and $A$, obtaining two sets:

$$E_Q = \{e_i^Q\}_{i=1}^n, \quad E_A = \{e_j^A\}_{j=1}^m. \tag{1}$$

$E_Q$ and $E_A$ denote the sets of entities extracted from the question and the answer, respectively, where $n$ and $m$ are the corresponding numbers of entities. These entities are then mapped to their corresponding nodes in the KG:

$$\hat{E}_Q = \{\hat{e}_i^Q \mid e_i^Q \mapsto \hat{e}_i^Q \in V\}, \quad \hat{E}_A = \{\hat{e}_j^A \mid e_j^A \mapsto \hat{e}_j^A \in V\}. \tag{2}$$

$\hat{E}_Q$ and $\hat{E}_A$ represent the sets of nodes in the KG corresponding to the extracted entities.

Table 1: Illustration of task categories at different difficulty levels. Basic tasks rely on short, direct reasoning in the KG (e.g., drug indication, biological process). Medium tasks involve longer, contextualized reasoning across intermediate entities (e.g., Off-label use, disease–protein associations, side effects). Hard tasks require multi-hop integration of heterogeneous concepts (e.g., contraindications, drug–drug interactions). Here, the blue node denotes the head entity in the question, and the orange node denotes the correct answer entity.

| Level | Task Category | Example | Illustration |
|---|---|---|---|
| **Basic** | Indication | Which **disease** can be treated with **Dalfampridine**? | |
| | Bioprocess | Which **biological process** is associated with **LIMS1**? | |
| **Medium** | Off-label Use | Which **drug** is used Off-label for **botulism**? | |
| | Disease-Protein | Which **protein** is associated with **Cutrarino triad**? | |
| | Side effect | What is a known **side effect** of **Flurbiprofen**? | |
| **Hard** | Contraindication | Which **disease** is contraindication for **Nitrogen**? | |
| | Drug Drug Interaction | Which **drug** has a drug drug interaction with **Piritrexim**? | |

**Path definition and extraction.** Reasoning paths are instantiated from a predefined set of abstract templates, denoted as $\mathbb{T} = \{\mathcal{T}_1, \mathcal{T}_2, \ldots, \mathcal{T}_N\}$. Each template $\mathcal{T}_i \in \mathbb{T}$ encodes a specific topological structure of entity and relation transitions, designed to link a question entity $u \in \hat{E}_Q$ to a candidate answer entity $v \in \hat{E}_A$. These templates accommodate a spectrum of structural complexities, ranging from simple linear chains to intricate non-linear graphs. To illustrate the diversity of these templates, we define three fundamental structural patterns:

(i) *Linear chain structures*, where entities are connected by a single sequential relation chain:

$$\pi_{\text{lin}}(u, v) = (u = v_0 \xrightarrow{r_1} v_1 \xrightarrow{r_2} v_2 \cdots \xrightarrow{r_d} v_d = v). \qquad (3)$$

(ii) *Divergent structures*, where multiple reasoning trajectories emanate from a common entity:

$$\pi_{\text{div}}(u, v) = \left\{ (u = v_0 \xrightarrow{r_1} v_1), \ (v_0 \xrightarrow{r_2} v_2 \xrightarrow{r_3} \cdots \xrightarrow{r_d} v_d = v) \right\}. \qquad (4)$$

(iii) *Convergent structures*, where distinct relational branches terminate at a common entity:

$$\pi_{\text{con}}(u, v) = \left\{ (u = v_0 \xrightarrow{r_1} v_1 \xrightarrow{r_2} \cdots v_d = v), \ (v_0 \xrightarrow{r_3} v_2 \xrightarrow{r_4} \cdots v_d = v) \right\}. \qquad (5)$$

In this formulation, $d$ denotes the overall reasoning complexity of a path, which jointly captures (i) the *depth*, corresponding to the maximal length of a relational chain, and (ii) the *breadth*, reflecting the number of admissible branches within the structure.

For a given template $\mathcal{T} \in \{\mathrm{lin}, \mathrm{div}, \mathrm{con}\}$, we define the set of all concrete reasoning paths derived from $\mathcal{T}$ as

$$\mathrm{Inst}(\mathcal{T}; u, v) \triangleq \pi(u, v) \mid \pi(u, v) \models \mathcal{T}. \tag{6}$$

Accordingly, the set of all instantiated paths connecting the question $Q$ and candidate answers $A$ can be formally expressed as

$$\mathcal{P}(Q, A) = \bigcup_{u \in \hat{E}_Q, \, v \in \hat{E}_A} \left\{ \pi(u, v) \mid \pi(u, v) \in \mathrm{Inst}(\mathcal{T}; u, v), \ \mathcal{T} \in \mathbb{T}, \ v_0 = u, \ v_{|\pi|} = v \right\}, \tag{7}$$

which collects all admissible paths derived from any template in $\mathbb{T}$ connecting entities in $Q$ to candidate answers in $A$.

**Task difficulty characterized by hop depth.** We define task difficulty through the complexity metric $d$, which aggregates two key factors: the maximal depth of the relational chain and the breadth of its structural branches. Greater depths correspond to broader knowledge scopes and higher degrees of informational fragmentation, thereby requiring more demanding semantic integration and logical inference. Accordingly, task difficulty is stratified into discrete levels as a function of $d$ (Eq. 8).Table 1 further illustrates the mapping between representative task categories and their associated depth ranges.

$$\text{Task Difficulty} = \begin{cases} \text{Basic (e.g., Indication, Bioprocess),} & d \leq 5, \\ \text{Medium (e.g., Off-label use, Disease–Protein, Side effect),} & 6 \leq d \leq 7, \\ \text{Hard (e.g., Contraindication, Drug Drug Interaction),} & d \geq 8. \end{cases} \tag{8}$$

In the next stage, these reasoning paths serve as the factual and structural foundation for guiding CoT generation.

## 3.2 CoT Generation and Pruning

Building upon the retrieved reasoning paths $P$, the next step is to generate a fact-grounded CoT that faithfully reflects the reasoning process underlying the question–answer pair. Each path $\pi \in \mathcal{P}$ provides a structured sequence of entity and relation transitions that can serve as a scaffold for the CoT, ensuring that the reasoning process is grounded in valid biomedical knowledge.

**CoT generation.** We define a prompt constructor $\Phi$ that incorporates the question $Q$, the correct answer $A$, the reasoning paths $\mathcal{P}$. The details of this prompt constructor is illustrated in Figure 6 of Appendix A.3. Given this structured input, the LLM produces a reasoning chain, which we denote as:

$$C = \mathrm{LLM}(\Phi(Q, A, \mathcal{P})). \tag{9}$$

Here, $C$ represents the initial CoT generated by the LLM. The generation process is guided by KG-derived evidence, ensuring that the generated CoT remains aligned with established biomedical knowledge and avoids unsupported associations.

**CoT pruning.** To enhance the clarity of the factually-grounded CoT($C$), which may contain redundant steps or tangential details, we define a pruning prompt constructor $\Psi$ that takes as input the question $Q$ and the initial reasoning chain $C$. The details of $\Psi$ is illustrated in Figure 7 of Appendix A.3. This prompt instructs the LLM to refine $C$ by removing unnecessary reasoning steps and preserving only the essential logical chain leading to the answer:

$$C_{\mathrm{pruned}} = \mathrm{LLM}(\Psi(Q, C)). \tag{10}$$

Here, $C_{\mathrm{pruned}}$ represents the refined reasoning chain that retains only the essential logical steps. Through this prompt-based pruning strategy, the reasoning process becomes more concise and interpretable while maintaining accuracy. A concrete example is provided in Figure 5 of Appendix A.3.

In summary, by leveraging KG-derived paths $\mathcal{P}$, this two-stage process generates a concise and evidence-grounded reasoning chain $C_{\mathrm{pruned}}$, achieving both factual correctness and interpretability.

### 3.3 SUPERVISED FINE-TUNING AND REINFORCEMENT LEARNING

After building the reasoning dataset, we conduct supervised finetuning and reinforcement learning. Through the SFT stage, the base model (Qwen3-4B and Qwen3-8B) are trained to follow KG-guided templates, generate coherent multi-step reasoning, and ensure output validity for complex biomolecular questions. This stage effectively mitigates reward sparsity, establishing a robust foundation for the model to efficiently refine its reasoning capabilities during the reinforcement learning phase.

For the reinforcement learning phase, we adopt Group Relative Policy Optimization (GRPO) (Shao et al., 2024), which leverages intra-group advantage estimation to eliminate the need for a separate critic network. The training process for each input $x$ involves sampling $G$ candidate responses from the current policy and optimizing it using a clipped objective function with group-normalized advantages (the detailed formulation is provided in Appendix A.1).

To simultaneously promote structured reasoning and factual accuracy, we design a composite reward function that enforces both the output format and the correctness. Each response is required to follow a reasoning template, containing intermediate reasoning in `<think>...</think>` and the final answer in `<answer>...</answer>`. The reward is composed of two parts:

$$R_{\text{format}} = \begin{cases} 1, & \text{if format is valid,} \\ 0, & \text{otherwise,} \end{cases} \qquad R_{\text{answer}} = \begin{cases} 5, & \text{if the predicted answer is correct,} \\ 0, & \text{otherwise,} \end{cases} \tag{11}$$

and the total reward is $R_{\text{reward}} = R_{\text{format}} + R_{\text{answer}}$. This reward structure ensures that answer correctness grants a primary reward of 5, while format adherence provides an additional reward of 1. In this way, these incentives promote both syntactic validity and semantic accuracy, guiding the model toward reliable reasoning and outputs.

## 4 EXPERIMENT AND RESULTS

### 4.1 EXPERIMENTAL SETUP

**Dataset and Benchmark** Existing biomedical QA benchmarks (e.g., BioASQ(Krithara et al., 2023), BiomixQA(Soman et al., 2024)) remain limited in scope: they are relatively small and lack explicit annotations for multi-hop reasoning over structured knowledge. These shortcomings restrict their ability to evaluate the deep reasoning capacities of the models. In contrast, we introduce PrimeKGQA, a benchmark constructed from PrimeKG (Chandak et al., 2023). QA pairs are generated using a template-based approach: Each question is formed from a head–relation pair, with the corresponding tail entity serving as the answer (Figure 2a). The tasks cover diverse biomedical categories(e.g., diseases, drugs, genes, pathways) and exhibit varying levels of reasoning difficulty. In particular, the path length in PrimeKG naturally reflects the complexity of the reasoning required to answer a question (Table 1). To prevent data leakage, the dataset is partitioned by head entities, ensuring no overlap between training and test sets. In total, PrimeKGQA consists of 6,710 QA pairs, with 3,500 for supervised fine-tuning, 1,500 for reinforcement learning, and 1,710 for evaluation. Detailed statistics of the test data are provided in the Appendix A.2. The QA pairs are in the form of single-choice questions with four options, and the evaluation metric is accuracy.

**Baselines** For the empirical validation of our proposed data and training methodology, we conduct a comparative analysis against a comprehensive set of baselines. To enable a structured comparison, these baselines are grouped into three categories: (1) Open-source model Qwen3-4B and Qwen3-8B (Yang et al., 2025) (base model baseline); (2) Closed-source models, including GPT-4o-mini, GPT-4o (Hurst et al., 2024), and Gemini 2.5 Pro (Comanici et al., 2025); (3) Reasoning-oriented models, such as o1 (Jaech et al., 2024), o3-mini, and Deepseek-R1 (Guo et al., 2025).

**Training and Inference** We perform further training on the Qwen3-4B and Qwen3-8B models (Yang et al., 2025). The training process follows a two-stage paradigm, consisting of full finetuning and reinforcement learning(RL), using the PrimeKGQA dataset (see Section 4.1). In the finetuning stage, the model is trained for 4 epochs with a learning rate of $1.0 \times 10^{-5}$, a per-device batch size of 2, and a maximum input length of 4096 tokens. For the subsequent RL phase, we transition to a parameter-efficient approach, employing Low-Rank Adaptation (LoRA) with a rank of 32. The model undergoes one epoch of training with a learning rate of $1.0 \times 10^{-6}$, a per-device batch

Table 2: Performance comparison of Open-source, Closed-source, Reasoning-oriented models, and our proposed model on the PrimeKGQA benchmark test set. The table shows accuracy scores across task categories of varying difficulty levels. Higher values indicate better performance.

| Type | Model | Basic | | | Medium | | | | Hard | | | All Avg. |
|------|-------|-------|---|---|--------|---|---|---|------|---|---|----------|
| | | Indi-cation | Bio-process | Avg. | Off-label use | Disease-Protein | Side effect | Avg. | Contra-indication | DDI | Avg. | |
| Open Source | Qwen3-4B | 0.768 | 0.523 | 0.606 | 0.710 | 0.573 | 0.616 | 0.618 | 0.438 | 0.503 | 0.473 | 0.568 |
| | Qwen3-8B | 0.845 | 0.653 | 0.749 | 0.671 | 0.600 | 0.516 | 0.586 | 0.452 | 0.537 | 0.498 | 0.601 |
| Closed Source | GPT-4o-mini | 0.903 | 0.740 | 0.796 | 0.903 | 0.713 | 0.844 | 0.801 | 0.564 | 0.683 | 0.625 | 0.743 |
| | Gemini 2.5 Pro | 0.910 | 0.850 | 0.880 | 0.710 | 0.803 | 0.804 | 0.783 | 0.664 | 0.647 | 0.655 | 0.768 |
| | GPT-4o | 0.890 | 0.873 | 0.879 | 0.897 | 0.807 | 0.844 | 0.840 | 0.676 | 0.623 | 0.650 | 0.789 |
| Reasoning oriented | Deepseek-R1 | 0.897 | 0.777 | 0.837 | 0.839 | 0.750 | 0.803 | 0.788 | 0.532 | 0.625 | 0.583 | 0.735 |
| | o3-mini | 0.897 | 0.877 | 0.884 | 0.852 | **0.857** | 0.772 | 0.826 | 0.580 | 0.663 | 0.625 | 0.777 |
| | o1 | **0.936** | **0.893** | **0.908** | 0.832 | 0.840 | 0.844 | 0.840 | 0.696 | **0.703** | 0.700 | 0.813 |
| Ours | **Bio-KCoT(4B)** | 0.897 | 0.773 | 0.815 | 0.890 | 0.743 | **0.912** | 0.835 | 0.804 | 0.610 | 0.698 | 0.786 |
| | **Bio-KCoT(8B)** | 0.901 | 0.807 | 0.839 | **0.930** | 0.793 | 0.908 | **0.864** | **0.848** | 0.643 | **0.736** | **0.816** |

size of 4, and a maximum input length of 4096 tokens. During inference, all models are prompted to generate CoT style responses.

## 4.2 MAIN RESULTS

Table 2 reports the experimental results on the PrimeKGQA benchmark. On basic-level tasks, advanced closed-source LLMs still maintain a leading position, primarily due to their extensive parameter knowledge and strong factual memory capabilities. However, as task difficulty increases, this advantage gradually diminishes and is eventually surpassed. On medium-level tasks, the Bio-KCoT 8B model achieves an average score of 0.864, outperforming both o1 and GPT-4o, which each scored 0.840. Specifically, in the Off-label use subtask, Bio-KCoT 8B achieves 0.930, significantly higher than GPT-4o's 0.897. In the side effect subtask, the 4B and 8B versions of Bio-KCoT achieve 0.912 and 0.908, respectively, far exceeding the 0.844 scores of GPT-4o and o1.

On the most complex hard-level tasks, Bio-KCoT 8B continues to lead with an overall score of 0.736. Its advantage is more pronounced in the contraindication subtask, where it achieves 0.848, substantially ahead of o1 at 0.696. The experiments also show that Bio-KCoT's performance improves with increased model size, demonstrating a favorable scale-up property. These results indicate that, with careful framework design and the incorporation of domain knowledge, even models with relatively small parameter counts can achieve performance on complex biomedical tasks that matches or exceeds that of some closed-source large models.

In addition, we observe that the performance of most baseline methods declines progressively from the basic to the medium and hard levels as task difficulty increases. In contrast, our method performs better on medium tasks than on basic ones, and its results on hard tasks remain comparable to those on the basic level. This suggests that the incorporation of long chains of thought becomes particularly advantageous as task complexity increases, enabling the model to leverage reasoning chains more effectively when shallow memorization is insufficient. By generating knowledge-guided reasoning chains, the model can gradually unfold intermediate steps and integrate structured knowledge with semantic information, thereby addressing high-complexity problems more effectively. To identify limitations for future improvement, see the error analysis in the Appendix A.5.

## 4.3 GENERALIZATION TO OTHER BENCHMARKS

To systematically evaluate the cross-task generalization ability of our method, we further conduct experiments on three external biomolecular question-answering datasets (results shown in Figure 3a). We directly evaluate the model on these out-of-distribution datasets without any further training. These benchmarks span different dimensions of biomolecular challenges: BioASQ (Krithara et al., 2023) reflects real expert information needs based on MEDLINE; BiomixQA (Soman et al., 2024) consists of multiple-choice questions derived from SPOKE disease–gene associations to assess com-

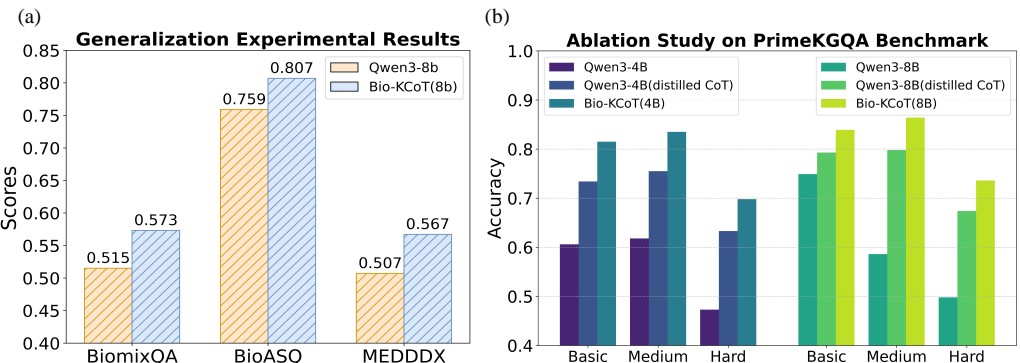

Figure 3: (a) Generalization results on additional biomedical benchmarks(BiomixQA, BioASQ, and MEDDDX). (b) Ablation study on the PrimeKGQA benchmark against the distilled CoT baseline.

plex relational reasoning; and MEDDDX (Su et al., 2024) leverages STaRK-Prime (Wu et al., 2024) to construct closely related distractors for evaluating fine-grained semantic discrimination.

Across these mutually distinct tasks, Bio-KCoT(8B) consistently achieves significant improvements over the base model Qwen3-8B. Notably, Bio-KCoT requires neither task-specific training nor complete external retrieval. Instead, our approach leverages knowledge graph-guided long chains of thought to effectively activate the model's intrinsic reasoning capacity. This mechanism allows the model to derive reliable answers without depending on fully comprehensive external knowledge, while stably transferring its capabilities to new tasks and domains. These results demonstrate that long CoT not only enhances reasoning performance on in-domain tasks but also exhibits stronger cross-task generalization, laying the foundation for deployment in complex real-world scenarios.

## 4.4 ABLATION STUDY

We conduct ablation studies to analyze the effectiveness of the proposed knowledge-guided long chain-of-thought generation method. Specifically, we compare the following approaches: (1) the base model Qwen3, on which all our methods are built; (2) Qwen3 (distilled CoT), where the supervised fine-tuning stage employs simple reasoning chains distilled from LLMs rather than knowledge-guided ones; and (3) our proposed Bio-KCoT method. All three approaches are evaluated with models of 4B and 8B parameters.

The experimental results are shown in Figure 3b. Analysis of the experimental results reveals a distinct performance hierarchy that validates our proposed approach. While applying a two-stage training paradigm (SFT and RL) with distilled CoT data yields a significant performance uplift compared to the original Qwen3 base model, our model, trained with Bio-KCoT, consistently and substantially outperforms this distilled CoT counterpart across all task difficulties. Crucially, the superior performance of our model validates the efficacy of our proposed KG-guided reasoning framework, demonstrating its advanced capability in navigating complex, multi-step problems. See the Appendix A.4 for detailed results.

## 5 CONCLUSION

We introduce Bio-KCoT, a knowledge-augmented long chain-of-thought reasoning framework for complex biomolecular problems. Our approach tackles the issues of unreliable knowledge and logical errors in large language models by using structured knowledge to guide reasoning, without depending on complete external information. Specifically, Bio-KCoT employs a knowledge graph–guided path search and pruning strategy to construct biologically meaningful and logically coherent reasoning chains. Further, we introduce PrimeKGQA, a benchmark dataset covering diverse biomolecular task types with various difficulties. Experimental results demonstrate that Bio-KCoT improves performance on this benchmark, particularly on hard-level questions, and maintains strong generalization on out-of-distribution datasets without additional training.

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

# A APPENDIX

## A.1 DETAILED GRPO OBJECTIVE AND IMPLEMENTATION

For a training input $x$, we sample $G$ candidate responses $\{y_i\}_{i=1}^G$ from the old policy $\pi_{\theta_{\text{old}}}$, and optimize the updated policy $\pi_\theta$ via the following clipped surrogate objective:

$$\mathcal{J}(\theta) = \mathbb{E}_{x\sim\mathcal{D},\ \{y_i\}_{i=1}^G\sim\pi_{\theta_{\text{old}}}(\cdot|x)} \frac{1}{G}\sum_{i=1}^G \Big[ \min\big(p_i(\theta)A_i,\ \text{clip}(p_i(\theta),\ 1-\epsilon,\ 1+\epsilon)\,A_i\big)$$
$$-\ \beta\,\mathbb{D}_{\text{KL}}\big(\pi_\theta \,\|\, \pi_{\theta_{\text{ref}}}\big)\Big], \tag{12}$$

where $p_i = \dfrac{\pi_\theta(y_i \mid x)}{\pi_{\theta_{\text{old}}}(y_i \mid x)}$ is the probability ratio.

The normalized advantage $A_i$ is computed within the sampled group:

$$A_i = \frac{r_i - \text{mean}(\{r_j\}_{j=1}^G)}{\text{std}(\{r_j\}_{j=1}^G)}\ , \tag{13}$$

with $r_i$ the reward of the $i$-th response. The KL penalty term $\mathbb{D}_{\text{KL}}$ stabilizes training against the reference policy $\pi_{\theta_{\text{ref}}}$, where $\epsilon$ is the clipping threshold and $\beta$ the penalty weight.

Here, we provide a detailed breakdown of the components in the objective function. In this formulation, $\mathcal{D}$ is the distribution of training prompts $x$, and $G$ is the number of candidate responses sampled per prompt. The policy being optimized is $\pi_\theta$, with parameters $\theta$, while $\pi_{\theta_{\text{old}}}$ is a fixed, older version of the policy used for sampling. The reference policy, $\pi_{\theta_{\text{ref}}}$, is used to regularize $\pi_\theta$. The term $p_i(\theta)$ is the importance sampling ratio for the $i$-th response, and $A_i$ is the normalized advantage of that response, calculated using the scalar reward $r_i$. The $\text{clip}(\cdot)$ function constrains the probability ratio to stabilize training. The Kullback-Leibler (KL) divergence, $\mathbb{D}_{\text{KL}}(\pi_\theta \,\|\, \pi_{\theta_{\text{ref}}})$, acts as a penalty term, with its strength controlled by the coefficient $\beta$. Finally, $\epsilon$ is the clipping hyperparameter for the surrogate objective.

## A.2 EVALUATION DATASETS

The distribution of the PrimeKGQA test dataset across various tasks is presented in Figure 4. The number of test samples for each task is proportional to the volume of corresponding data in the source KG, thus aligning the evaluation with the inherent data distribution of the KG. Table 3 presents a quantitative overview of our test data, illustrating the statistical distribution of samples between the primary test set and the generalization benchmarks. The summary specifically delineates the volume and variety of type of questions within each data set, establishing a clear and comprehensive foundation for our performance evaluation.

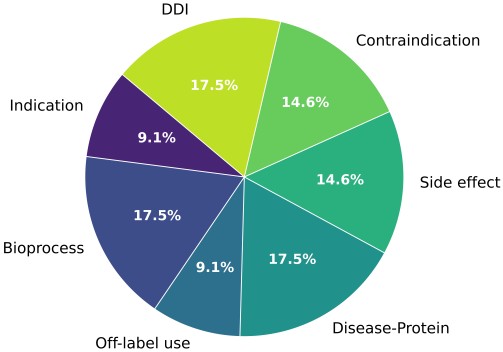

Figure 4: The distribution of the PrimeKGQA test dataset across various tasks.

Table 3: Comprehensive statistics for our evaluation data, detailing the distribution of question types and counts across the primary test set and generalization benchmarks.

| Dataset | Question Type | Number |
|---|---|---|
| PrimeKGQA | MCQ | 1710 |
| BiomixQA | T/F Question | 498 |
| BioASQ | MCQ | 246 |
| MEDDDX | MCQ | 245 |

## A.3 CASE STUDY AND RELATED PROMPTS

To provide a concrete illustration of our methodology, we present a detailed case study (Figure 5). The process begins with the initial problem, from which we extract a relevant reasoning path within the KG. This path serves as the foundation for generating an initial, verbose response using our CoT generation prompt (Figure 6). Subsequently, this preliminary answer undergoes a refinement phase, where a specialized CoT pruning prompt is applied to revise the essential information and produce the final, concise answer (Figure 7). Throughout these figures, we use color-coding to highlight the transformation. Specifically, the text marked in red indicates superfluous content that deviates from a true inferential path. Such content is revised because it is either superfluous or fails to demonstrate true reasoning. This failure is evident when the model abandons the step-by-step derivation, instead opting to provide external facts directly or flagging its input with terms like "Additional knowledge". In contrast, the content marked in green represents the final, polished output, which is composed of the valid inferential chain that has been preserved and, in some cases, augmented with further elaborations to enhance logical coherence and completeness.

## A.4 DETAILED RESULTS

Our experimental results comprehensively validate the effectiveness and robustness of the proposed Bio-KCoT framework. The ablation study, detailed in Table 4, demonstrates that our method consistently and substantially outperforms both the base model and a strong distilled CoT baseline across all task categories and difficulty levels on the PrimeKGQA benchmark. This highlights the significant advantage of integrating structured knowledge into the reasoning process. Furthermore, as shown in Table 5, the performance gains are not confined to our primary dataset; the model maintains its superiority on several external biomedical benchmarks. This confirms that the enhanced reasoning capabilities generalize effectively to unseen data and diverse problem types, establishing Bio-KCoT as a powerful and broadly applicable framework for complex biomedical question answering.

Table 4: Ablation study on PrimeKGQA. Experiments are performed across all tasks with the 4B and 8B model variants, in comparison with the distilled CoT baseline.

| Model | Basic | | | Medium | | | | Hard | | | All Avg. |
|---|---|---|---|---|---|---|---|---|---|---|---|
| | Indi-cation | Bio-process | Avg. | Off-label use | Disease-Protein | Side effect | Avg. | Contra-indication | DDI | Avg. | |
| Qwen3-4B | 0.768 | 0.523 | 0.606 | 0.710 | 0.573 | 0.616 | 0.618 | 0.438 | 0.503 | 0.473 | 0.568 |
| Qwen3-4B (distilled CoT) | 0.819 | 0.690 | 0.734 | 0.800 | 0.687 | 0.808 | 0.755 | 0.720 | 0.560 | 0.633 | 0.710 |
| **Bio-KCoT(4B)** | 0.897 | 0.773 | 0.815 | 0.890 | 0.743 | **0.912** | 0.835 | 0.804 | 0.610 | 0.698 | 0.786 |
| Qwen3-8B | 0.845 | 0.653 | 0.749 | 0.671 | 0.600 | 0.516 | 0.586 | 0.452 | 0.537 | 0.498 | 0.601 |
| Qwen3-8B (distilled CoT) | 0.869 | 0.753 | 0.793 | 0.851 | 0.747 | 0.828 | 0.798 | 0.768 | 0.597 | 0.674 | 0.773 |
| **Bio-KCoT(8B)** | **0.901** | **0.807** | **0.839** | **0.930** | **0.793** | 0.908 | **0.864** | **0.848** | **0.643** | **0.736** | **0.816** |

Table 5: Generalization study on three biomedical benchmarks (BiomixQA, BioASQ, and MED-DDX). Performance is evaluated using the 4B and 8B model variants.

| Model | BiomixQA | BioASQ | MEDDDX |
|---|---|---|---|
| Qwen3-4b | 0.472 | 0.659 | 0.426 |
| **Bio-KCoT(4b)** | 0.512(**+4.0**) | 0.727(**+6.8**) | 0.473(**+4.7**) |
| Qwen3-8b | 0.515 | 0.759 | 0.507 |
| **Bio-KCoT(8b)** | 0.573(**+5.8**) | 0.807(**+4.8**) | 0.567(**+6.0**) |

## A.5 ERROR ANALYSIS

While the Bio-KCoT framework demonstrates substantial advancements in reasoning performance, a systematic error analysis is essential for diagnosing its inherent limitations and guiding future it-

erations. This section presents a detailed error analysis of failures arising during the generation and execution of reasoning chains. Errors are traced from the final answers (highlighted in red) back through the CoT generation and execution steps (Figure 8). This process helps to illuminate the model's current limitations, and informing pathways for future refinement. The analysis indicates that most errors do not arise from flawed logical deduction but are instead attributable to the limited knowledge capacity of smaller models, which may lead to errors when recalling specific facts or specialized knowledge points. Consequently, even when the reasoning process is internally consistent, the final outputs can contain inaccuracies or misconceptions, highlighting the importance of combining reasoning analysis with knowledge verification in future iterations.

## B  ETHICS STATEMENT

This work complies with the ICLR Code of Ethics. Our study did not involve human participants or animal testing. All datasets, including PrimeKGQA, were obtained and used in accordance with relevant guidelines, ensuring respect for data usage policies and privacy considerations. We carefully mitigated potential biases and avoided discriminatory effects during the research process. No personal or identifiable information was utilized, and no experiments were carried out that might raise concerns regarding security or confidentiality. Throughout the study, we have upheld principles of transparency, responsibility, and research integrity.

## C  REPRODUCIBILITY STATEMENT

To promote transparency and ensure reproducibility, we have provided all relevant code in the Supplementary Material. The manuscript offers a comprehensive description of the experimental setup, including the training pipeline, model architectures, hyperparameter configurations. These materials are intended to enable independent verification of our findings and to support subsequent research that builds upon our methodology.

## D  THE USE OF LARGE LANGUAGE MODELS

Large Language Models were employed as versatile assistive tools in this research. They were primarily used to support and refine the preparation of this manuscript, including correcting grammar, enhancing clarity, and restructuring sentences to improve overall readability. LLMs also assisted in generating foundational scripts for data preprocessing tasks such as cleaning, formatting, and visualization, which were subsequently reviewed and adjusted by the authors to ensure correctness and consistency with research objectives.

In addition, LLM APIs were invoked during the CoT generation and pruning stages, where they contributed to producing candidate reasoning traces and improving their quality through refinement. Across all these applications, the outputs provided by LLMs were critically assessed and edited by the authors. At no stage did the models substitute for human intellectual contributions; rather, they served to accelerate and augment different phases of the research workflow.

**Question**

**Which disease can be treated with Lisinopril?**
A. qualitative or quantitative protein defects in neuromuscular diseases
B. intellectual disability-hypotonia-brachycephaly-pyloric stenosis-cryptorchidism syndrome
C. myocardial infarction
D. craniofrontonasal dysplasia-Poland anomaly syndrome

**Extracted path from KG**

**Lisinopril** targets **ACE** and **REN**.
**ACE** is involved in the pathway: **Metabolism of Angiotensinogen to Angiotensins**, which is part of the broader **Peptide hormone metabolism pathway**.
**ACE** is associated with a wide range of diseases, including neurological disorders (e.g., schizophrenia, Alzheimer disease, cerebellar degeneration, anxiety disorder, bipolar disorder, depression), cardiovascular diseases (e.g., coronary artery disease, heart failure, **myocardial infarction**, stroke, atrial fibrillation), kidney diseases (e.g., chronic kidney disease, diabetic nephropathy, renal tubular dysgenesis), various cancers (e.g., lung cancer, liver cancer, gastric cancer, prostate cancer, salivary gland adenocarcinoma), metabolic and systemic disorders (e.g., substance abuse, hypotensive disorder, fatty liver disease), autoimmune and infectious conditions (e.g., celiac disease, staphylococcal infection, severe acute respiratory syndrome), and other conditions such as autism spectrum disorder and drug psychosis.
**REN** is involved in the same pathway:**Metabolism of Angiotensinogen to Angiotensins**, under **Peptide hormone metabolism**.
**REN** is associated with diseases including neurological and psychiatric disorders (e.g., anxiety disorder, depression, bipolar disorder, dysthymic disorder, neurotic disorder), cardiovascular and renal conditions (e.g., congestive heart failure, **malignant hypertension**, kidney disease, nephrosis, renal tubular dysgenesis, interstitial nephritis), metabolic and systemic disorders (e.g., anemia, osteoporosis, substance withdrawal syndrome, hypotensive disorder), autoimmune and skin diseases (e.g., psoriasis, pustulosis palmaris et plantaris, parapsoriasis), and genetic disorders affecting electrolyte balance (e.g., Bartter disease, Liddle syndrome, hyperaldosteronism, familial juvenile hyperuricemic nephropathy).

**CoT Generation**

**Step1:** The question asks which disease can be treated with Lisinopril, a drug known to target ACE (angiotensin-converting enzyme) and REN (renin). Both of these proteins play central roles in the Renin-Angiotensin-Aldosterone System (RAAS), which regulates blood pressure, vascular tone, and fluid balance. The key pathway involved is the metabolism of angiotensinogen to angiotensins, a subset of peptide hormone metabolism. Dysregulation of this pathway often leads to cardiovascular and renal pathologies, including hypertension, heart failure, and myocardial infarction.
**Step2:** Knowing Lisinopril is an ACE inhibitor, I can now use the provided "Additional knowledge". ACE is involved in the "Metabolism of Angiotensinogen to Angiotensins" pathway. This is the core of the Renin-Angiotensin-Aldosterone System (RAAS), a critical hormonal cascade that regulates blood pressure and fluid balance. By inhibiting ACE, Lisinopril prevents the conversion of angiotensin I to angiotensin II. Angiotensin II is a potent vasoconstrictor (it narrows blood vessels) and stimulates the release of aldosterone, which causes the body to retain salt and water. Therefore, blocking its production leads to vasodilation (widening of blood vessels) and a reduction in blood volume, both of which lower blood pressure and reduce the workload on the heart.
**Step3:** Now, I can evaluate each option in the context of Lisinopril's mechanism: A. qualitative or quantitative protein defects in neuromuscular diseases: These are typically genetic disorders affecting muscle function, like muscular dystrophies. The RAAS pathway is not the primary driver of this pathology. An ACE inhibitor would not correct underlying protein defects. B. intellectual disability-hypotonia-brachycephaly-pyloric stenosis-cryptorchidism syndrome: This is a complex, multi-system congenital syndrome. Its cause is genetic, not related to the hormonal regulation of blood pressure. Treatment would be supportive, and Lisinopril would not address the root causes. C. myocardial infarction (MI): This is a heart attack, a major cardiovascular event. The "Additional knowledge" explicitly states that ACE is associated with... cardiovascular diseases (e.g., ... myocardial infarction). This is a direct link. Clinically, ACE inhibitors like Lisinopril are a cornerstone of post-MI therapy because they reduce the strain on the damaged heart, lower blood pressure, and help prevent negative

Figure 5: A detailed case study illustrating our three-stage methodology. The example demonstrates the transformation of a response, starting from the foundational KG path, to a verbose answer via CoT generation, and concluding with a concise, refined answer after CoT pruning.

structural changes (remodeling) in the heart muscle, thereby improving survival and reducing the risk of future heart failure. The drug's mechanism aligns perfectly with this indication. D. craniofrontonasal dysplasia-Poland anomaly syndrome: This is another congenital disorder involving developmental abnormalities of the head, face, and chest. It has no mechanistic connection to the RAAS pathway. **Step4**: Based on the logical chain, Lisinopril's identity as an ACE inhibitor directly links it to the RAAS pathway. The function of this pathway is central to cardiovascular health. Of the choices provided, only myocardial infarction is a cardiovascular disease where the therapeutic effects of ACE inhibition are not only plausible but are a standard and evidence-based treatment. The provided knowledge directly confirms the association between the drug's target (ACE) and this specific disease.

## CoT Pruning

**Step 1:** Alright, the question is about which disease Lisinopril can treat. Lisinopril—I recognize that name. It's an angiotensin-converting enzyme (ACE) inhibitor. ACE is a key enzyme in the renin-angiotensin-aldosterone system (RAAS). Lisinopril works by binding to the active site of ACE, which normally converts angiotensin I to angiotensin II. Angiotensin II is a potent vasoconstrictor and also stimulates aldosterone release, leading to sodium and water retention. By inhibiting ACE, Lisinopril reduces angiotensin II levels, causing vasodilation, lowering blood pressure, and decreasing fluid volume. But it also increases bradykinin because ACE breaks down bradykinin—so less degradation means more bradykinin, which promotes vasodilation through nitric oxide and prostaglandins. This makes me think Lisinopril is primarily used for cardiovascular conditions like hypertension or heart failure. Now, I need to match this mechanism to one of the diseases listed.

**Step 2:** Let me evaluate each option carefully, starting with A: "qualitative or quantitative protein defects in neuromuscular diseases." This sounds like disorders such as muscular dystrophy or myopathies, where there's a defect in proteins like dystrophin or components of the muscle membrane. These involve structural issues in muscle cells or neuromuscular junctions. Lisinopril affects systemic vascular tone and fluid balance through RAAS, but it doesn't target protein synthesis, repair genetic defects, or directly influence neuromuscular signaling. I recall that treatments for neuromuscular diseases focus on physical therapy, immunosuppressants, or gene therapies—not ACE inhibitors. So, this seems like a distractor; Lisinopril wouldn't address the core pathology here. I can rule out A confidently.

**Step 3:** Next, option B: "intellectual disability-hypotonia-brachycephaly-pyloric stenosis-cryptorchidism syndrome." That's a mouthful—it seems like a rare genetic syndrome, possibly linked to chromosomal deletions or mutations affecting neurodevelopment and physical features. Intellectual disability and hypotonia suggest brain involvement, while pyloric stenosis and cryptorchidism are structural anomalies. Lisinopril's mechanism is purely pharmacological, modulating blood pressure and cardiac load. It doesn't cross the blood-brain barrier effectively to affect cognition, nor can it correct congenital malformations like undescended testes or stomach narrowing. Genetic syndromes require targeted interventions, not broad-spectrum drugs like ACE inhibitors. I'm doubtful this could be correct; it feels unrelated to Lisinopril's known actions. So, B is likely not the answer.

**Step 4:** Option C: "myocardial infarction." This is a heart attack, where blood flow to part of the heart is blocked, causing tissue death. Post-MI, the RAAS gets hyperactivated as a stress response—angiotensin II drives inflammation, fibrosis, and ventricular remodeling, which worsens heart function over time. Lisinopril should counteract this by reducing angiotensin II, thus lowering afterload, improving coronary perfusion, and preventing pathological changes like wall thinning and dilation. It's a standard post-MI therapy to prevent heart failure. This seems plausible, but I should check the other option before concluding.

**Step 5:** Finally, option D: "craniofrontonasal dysplasia-Poland anomaly syndrome." This appears to be another genetic disorder, involving craniofacial and skeletal defects like fused sutures or missing chest muscles. Mutations in genes like EFNB1 disrupt cell migration during development. Lisinopril can't alter genetic coding or fix structural birth defects—it might even be risky in syndromes with potential renal or vascular abnormalities. Treatments involve surgery or specialized care, not RAAS modulation. I have no evidence linking ACE inhibitors to such conditions; this option seems like a distractor similar to B. So, D is incorrect.

**Step 6:** Now, confirming my hypothesis for C: Lisinopril's role in myocardial infarction isn't just theoretical—it's well-established. By inhibiting RAAS, it reduces infarct expansion, stabilizes the extracellular matrix, and lowers oxygen demand. The bradykinin buildup might also have cardioprotective effects. Conclusion: After eliminating distractors—A (neuromuscular, no RAAS link), B (genetic neurodevelopmental, not pharmacologically addressable), and D (genetic skeletal, requires surgical intervention)—only C, myocardial infarction, aligns with Lisinopril's mechanism and real-world clinical applications. Therefore, C is the correct answer.

Figure 5: (continued)

---

**The prompt for Generation**

You are given a single-choice biological reasoning question. Your task is to generate a detailed step-by-step reasoning process that can guide a model to select the correct answer.

Question: {*question*}
Options: {*options*}
Additional knowledge: {*p*}
True answer: {*true_answer*}

Now, write a natural, plausible reasoning process that could lead a knowledgeable person to choose the correct answers. Avoid simply referencing pre-established facts, and instead build your reasoning as a sequence of logical thoughts that connect concepts.
Begin by examining {*head_node_type*} biological basis, including key pathways, affected tissues, and pathological mechanisms. Then reason about which proteins among the options are likely to be functionally or mechanistically involved, based on their roles and interactions. Evaluate each option independently and identify all proteins that are plausible based on this reasoning.

Do not just state facts from the "Additional knowledge",integrate them into a coherent narrative, making the reasoning flow as though it arises naturally from your own deep biological understanding.

Finally, return your answer as a JSON dictionary with the reasoning in the field "cot_reasoning".
Output format: *{{"cot_reasoning": ["xxx"]}}*

Figure 6: The prompt structure used for CoT generation. This prompt guides the model to produce a detailed, step-by-step reasoning process that leads to an initial answer based on the extracted KG path.

---

**The prompt for Pruning**

Modify the scientific reasoning to meet the following requirements.
Please follow these instructions:

1. Keep the technical accuracy and depth of the original reasoning.
2. Delete the content that is not related to the question.
3. Add realistic thought processes: moments of doubt and elimination of distractors.
4. Ensure every answer option is explicitly considered and evaluated.

Here is an example: {*example question*}
Original output: {*example original_output*}
The model should rewrite like this: {*example rewrite_output*}

Now, try to rewrite the thinking process of the next question in the same way.
Question: {*question*}
Original output: {*original_output*}

【Output Format】
*<reasoning>\n ...\n</reasoning>*

Figure 7: The prompt structure used for CoT pruning. This prompt instructs the model to refine the initial, verbose reasoning by eliminating redundancies, correcting logical failures, and distilling the core inferential steps into a final, concise answer.

**Error cases**

**Question**

Which disease is contraindication for Carbenicillin?
A. non-suppurative otitis media
B. optic nerve glioma
C. perinatal necrotizing enterocolitis
D. pregnancy associated osteoporosis

**Error answer**

**Step 1:** Okay, the question is about contraindications for Carbenicillin. First, I need to recall what Carbenicillin is. It's a beta-lactam antibiotic, specifically a carboxypenicillin, used primarily against Gram-negative bacteria like Pseudomonas aeruginosa. Contraindications are conditions where the drug should not be used due to potential harm. I remember Carbenicillin has known risks, especially with prolonged use or in vulnerable populations. The big ones are nephrotoxicity (kidney damage) and neurotoxicity (like seizures or nerve issues). Also, since it's a penicillin derivative, there's a risk of allergic reactions, but that's more of a general caution than a specific contraindication.
**Step 2:** Now, I need to evaluate each disease option systematically. Let's take them one by one, thinking about mechanisms and potential conflicts.\n\nA. Non-suppurative otitis media: This is a type of middle ear infection without pus, often viral. Antibiotics like Carbenicillin might be used if bacterial, but is it a contraindication? I'm doubtful. Non-suppurative cases are usually viral, so antibiotics aren't first-line. Even if bacterial, Carbenicillin's risks (nephrotoxicity, neurotoxicity) aren't directly tied to ear infections. The main risks here are misuse leading to resistance or side effects, not a specific prohibition. This seems like a distractor\u2014it's not a core contraindication, so I can probably rule it out. B. Optic nerve glioma: This is a tumor in the optic nerve, often seen in children. Hmm, why might Carbenicillin be contraindicated here? I recall that beta-lactam antibiotics like Carbenicillin can cause neurotoxicity, including optic neuritis or nerve damage. If someone already has an optic nerve glioma, giving this drug could worsen the condition by exacerbating inflammation or toxicity in that nerve. That makes sense\u2014using a neurotoxic drug in a patient with pre-existing optic nerve damage could lead to vision loss or accelerated tumor progression. This feels plausible, but I should check the other options before deciding.\n\nC. Perinatal necrotizing enterocolitis: This is a serious intestinal condition in newborns. Carbenicillin is sometimes used in neonates for infections, but is it contraindicated? I'm skeptical. While antibiotics can contribute to NEC in some cases (e.g., by disrupting gut flora), Carbenicillin isn't specifically contraindicated. Its risks like nephrotoxicity might be more relevant in older patients than in neonates. Also, NEC management focuses on supportive care and avoiding certain drugs, but Carbenicillin isn't a typical no-go. This seems like another distractor\u2014no strong link to the drug's specific dangers.\n\nD. Pregnancy associated osteoporosis: This is bone loss in pregnant women, often due to hormonal changes. Carbenicillin is a penicillin, and penicillins can be used safely in pregnancy (like for bacterial infections), but they're not known to cause osteoporosis or worsen it. The drug's metabolism doesn't involve bone pathways, and there's no evidence it affects calcium or vitamin D. While all drugs should be used cautiously in pregnancy, this isn't a specific contraindication. I can eliminate this one\u2014it's unrelated.
**Step 3:** Going back to option B, I need to test the hypothesis. If optic nerve glioma is a contraindication, why? Carbenicillin's neurotoxicity can manifest as optic neuritis, which might mimic or worsen a glioma. In clinical practice, avoiding neurotoxic drugs in patients with pre-existing optic nerve pathology is prudent to prevent irreversible damage. For instance, in conditions like multiple sclerosis, neurotoxic drugs are avoided for similar reasons. This aligns with Carbenicillin's profile, making B the strongest candidate.
**Step 4:** Final confirmation. After eliminating A, C, and D, only B has a direct mechanistic link: Carbenicillin's potential to cause neurotoxicity could harm the optic nerve in a glioma patient. No other option has this level of risk association. So, the contraindication stands.

Figure 8: An illustrative example of the error analysis process for a failure case. The error is traced from the final answer back through the reasoning chain, revealing that the failure originates from a knowledge deficit rather than a flaw in logical deduction.

