# OpenReview forum: "Knowledge-Augmented Long-CoT Generation for Complex Biomolecular Reasoning"
_ICLR.cc/2026/Conference — Submitted to ICLR 2026_

### Official Review · Reviewer_A8FZ · 2025-10-28

**Soundness:** 2
**Presentation:** 2
**Contribution:** 3
**Rating:** 4
**Confidence:** 3

**Summary:**

This paper proposes **Bio-KCoT**, a knowledge-augmented long chain-of-thought framework that integrates knowledge graphs with LLM reasoning for complex biomolecular problems. It also introduces **PrimeKGQA**, a new benchmark designed to evaluate multi-hop biomedical reasoning.

**Strengths:**

1. Clear problem motivation: LLMs struggle with biologically grounded multi-step reasoning; the paper argues for structured KG guidance instead of pure retrieval completeness.

2. Clear and coherent methodology: The pipeline is well-structured, integrating KG-guided path retrieval, CoT generation, and SFT + RL training in a logical and effective way.

3. Meaningful benchmark contribution: The PrimeKGQA dataset is thoughtfully designed, with multi-level tasks that capture biologically relevant reasoning depth and enable systematic evaluation.

**Weaknesses:**

1. Limited novelty and conceptual contribution: The work essentially fine-tunes a small open model (Qwen) on a domain-specific KG reasoning task using SFT and GRPO. While technically sound, the novelty beyond applying existing RLVF-style training to this setting is not strongly articulated.

2. Main comparison lacks stronger baselines: The key experiment (Figure 3a) mainly compares Bio-KCoT with its own base model Qwen, which naturally benefits from 'in-benchmark' training. More direct comparisons with other reasoning-oriented or biomedical LLMs would better establish the model’s advantage and fairness.

3. Insufficient ablation analysis: The ablations are limited and leave key questions unanswered: (a) there is no analysis on how SFT and RL respectively contribute to performance or how their data ratios affect results; and (b) the comparison between Bio-KCoT and distilled CoT is only shown on one benchmark (Figure 3b), without testing across datasets. These gaps make it hard to assess the necessity and effectiveness of each pipeline component.

**Questions:**

The paper notes that QA pairs are generated from “head–relation pairs, with the corresponding tail entity serving as the answer.” It remains unclear how these head–relation pairs are selected from the knowledge graph and how question quality is ensured. Could the authors clarify the selection and filtering process, and whether any human validation was used to confirm that the generated questions are meaningful rather than trivial or noisy?

---

> ### Author Response · Authors · 2025-11-23
> **Summary of our response**
>
> We thank the reviewer for the constructive feedback and for recognizing the clear motivation, coherent methodology, and the value of our PrimeKGQA benchmark. We appreciate the opportunity to clarify the novelty, baselines, and ablation studies.
>
> ### **W1: Novelty and Conceptual Contribution**
>
> We clarify that our core contribution is data-centric: we leverage KG guidance to teach the model structured thinking, rather than simply applying SFT and RL. The main bottleneck in biomedicine is not the training algorithm, but the lack of reliable reasoning chains, which often leads to hallucinations. Bio-KCoT addresses this by transforming static knowledge graph paths into clear, logical reasoning steps. By rigorously grounding generation in the KG and applying our novel pruning step, we filter out noise and ensure accuracy. This process effectively enables the model to internalize structured biological logic and causal dependencies—a capability that standard fine-tuning on unstructured text cannot achieve.
>
> ### **W2: Baselines and Comparisons**
>
> To address the need for stronger baselines, we conducted additional experiments with  Meditron-7B, BioMistral-7B and OpenBioLLM-8B on OOD benchmarks. Surprisingly, these domain-specific models consistently underperformed the generalist Qwen3-8B baseline, indicating that domain adaptive pre-training does not automatically translate to complex multi-hop reasoning skills.
>
> In contrast, **Bio-KCoT (8B)** achieved superior results across all metrics (e.g., $0.573$ on BiomixQA), significantly outperforming both the generalist base and specialized biomedical models. This validates that our framework's advantage stems from KG-guided structured reasoning rather than the base model's capability or simple knowledge injection. We will add these comparisons to Figure 3a.
>
> | Model | BiomixQA | BioASQ | MEDDDX |
> | :--- | :---: | :---: | :---: |
> | Meditron-7B | $0.224$ | $0.564$ | $0.282$ |
> | BioMistral-7B | $0.427$ | $0.584$ | $0.405$ |
> | OpenBioLLM-8B | $0.451$ | $0.438$ | $0.441$ |
> | Qwen3-8B | $0.515$ | $0.759$ | $0.507$ |
> | **Bio-KCoT (8B)** | $\mathbf{0.573}$ | $\mathbf{0.807}$ | $\mathbf{0.567}$ |
>
> ### **W3: Ablation Analysis**
>
> We have conducted detailed ablations on 4B/8B models using 0%, 50%, and 100% RL data ratios to isolate the contributions of SFT and RL. The results demonstrate distinct roles:
> * **SFT** establishes a robust foundation, with the SFT-only 8B model achieving $0.798$, a massive leap over the base Qwen3-8B ($0.601$).
> * **RL** acts as a critical refiner for complex reasoning, driving a consistent performance increase correlated with data coverage (e.g., 8B overall score rises to $0.816$ at 100% RL).
> * Notably, RL contributes most significantly to "Hard" tasks (e.g., Contraindication/DDI), where the 8B model improves from $0.718$ to $0.736$, confirming that while SFT instills knowledge formats, RL is indispensable for minimizing logical errors in multi-hop scenarios.
>
> | Model | Indication | Bioprocess | Avg. (Basic) | Off-label | Disease-Protein | Side effect | Avg. (Med) | Contra-indication | DDI | Avg. (Hard) | **All Avg.** |
> | :--- | :---: | :---: | :---: | :---: | :---: | :---: | :---: | :---: | :---: | :---: | :---: |
> | SFT+0%RL(4B) | 0.870 | 0.763 | 0.799 | 0.878 | 0.717 | 0.906 | 0.819 | 0.784 | 0.602 | 0.685 | 0.771 |
> | SFT+50%RL(4B) | 0.874 | 0.770 | 0.805 | 0.893 | 0.737 | 0.896 | 0.828 | 0.788 | 0.610 | 0.691 | 0.778 |
> | SFT+100%RL(4B) | 0.897 | 0.773 | 0.815 | 0.890 | 0.743 | 0.912 | 0.835 | 0.804 | 0.610 | 0.698 | 0.786 |
> | SFT+0%RL(8B) | 0.909 | 0.777 | 0.822 | 0.909 | 0.783 | 0.880 | 0.845 | 0.824 | 0.630 | 0.718 | 0.798 |
> | SFT+50%RL(8B) | 0.890 | 0.783 | 0.819 | 0.916 | 0.780 | 0.904 | 0.854 | 0.834 | 0.653 | 0.735 | 0.806 |
> | **SFT+100%RL(8B)** | **0.901** | **0.807** | **0.839** | **0.930** | **0.793** | **0.908** | **0.864** | **0.848** | **0.643** | **0.736** | **0.816** |
>
> ### **Q: QA Generation and Quality Control**
>
> We ensure that the generated questions are biologically meaningful and non-trivial through a rigorous, multi-layered quality assurance pipeline:
> 1.  **Selection:** We utilize PrimeKG, a scientifically curated knowledge graph, as our ground truth, applying strict rule-based filtering to retain only biologically significant pathways rather than trivial connections.
> 2.  **Hard Negative Strategy:** To guarantee task difficulty and quality, we employ a hard negative strategy where distractors are strictly selected from entities of the same type as the correct answer, thereby testing the model's ability to distinguish true causal links from mere co-occurrence.
> 3.  **Prevention of Data Leakage:** We rigorously prevent data leakage by partitioning the dataset based on **Head Entities** to enforce generalizable reasoning. Additionally, we conducted manual expert reviews of the generation templates to confirm that the resulting QA logic is scientifically sound.
>
> We once again thank the reviewer for their time and valuable comments.

---

> > ### Comment · Reviewer_A8FZ · 2025-11-25
> >
> > Thank you for your thoughtful and detailed rebuttal, which has addressed several of my concerns effectively. I appreciate the additional comparisons with domain-specific models and the insights into your ablation analysis, which help clarify the strengths of your work. However, I would like to explore two points further.
> > 1. Choice of SFT+RL Pipeline:
> > I apologize for not being clear in my earlier question. Could you explain why the SFT+RL pipeline was chosen over using purely SFT, or why using all 5000 data points for SFT (without RL) wasn’t tested? Additionally, it would be helpful to understand if other SFT:RL data splits were considered and why the chosen split was optimal.
> > 2. QA Generation:
> > Regarding the "strict rule-based filtering" for biologically significant pathways, could you clarify how relations are selected? I understand this might be challenging to explain in detail, but any further insight into the process would be helpful.
> >
> > Once again, I appreciate your effort in addressing the concerns raised.

---

### Official Review · Reviewer_8tFo · 2025-11-01

**Soundness:** 2
**Presentation:** 2
**Contribution:** 2
**Rating:** 4
**Confidence:** 4

**Summary:**

This paper proposes Bio-KCoT, a framework that integrates knowledge graph-guided reasoning with large language models for biomolecular question answering. The approach constructs reasoning paths from knowledge graphs, generates long chain-of-thought responses, and uses a two-stage training process (SFT + GRPO). The authors also introduce PrimeKGQA, a benchmark derived from the PrimeKG knowledge graph.

**Strengths:**

1. Important Problem Domain: Biomolecular reasoning is a critical application area where factual accuracy and structured knowledge integration are essential. The motivation to address hallucinations and logical inconsistencies in this high-stakes domain is well-founded.

2. Comprehensive Framework Design: The three-stage approach (KG path extraction → CoT generation → pruning) is systematic and well-motivated. The integration of structured knowledge with long-form reasoning addresses real limitations of current LLMs.

3. Comprehensive results: show improvement not only on in domain benchmark, but on domains requiring generalization.

**Weaknesses:**

1. My main concern is about benchmark construction:

PrimeKGQA is constructed from PrimeKG which integrates 20 high-quality biomedical resources to describe 17,080 diseases with 4,050,249 relationships, but the benchmark creation is template-based where questions are automatically generated from head-relation-tail triples. This raises several concerns:

- Artificial Task Design: The problems may not truly require complex KG reasoning or long CoT. Many could potentially be answered with parameterized knowledge without reasoning.
- Limited Reasoning Depth: The "difficulty" categorization based on path length (d ≤ 5 for basic, 6-7 for medium, ≥8 for hard) may not reflect genuine reasoning complexity.
- Lack of novelty in terms of tasks: these are all basic drug discovery tasks common in bio and molecular domains. It would be more interesting that these questions are beyond simple knowledge-based QA and have more authentic scenarios, such as research questions.


2. Knowledge Graph Dependency: The approach is fundamentally limited by KG coverage and quality. Many biomedical questions may require reasoning beyond what's explicitly encoded in structured KGs. Can this method be extended into general retrieval of related knowledge, including online search and textual databases. Also, it lacks comparison with DeepResearch style models.

**Questions:**

1. Path Quality: How do you ensure that the extracted KG paths are actually relevant for reasoning rather than just providing factual connections? e.g. for drug-drug interaction, most of the connections after a few hops may be weak and not logical.

2. Reasoning vs. Retrieval: Many of the tasks shown (Table 1) appear to be factual lookups (e.g., "Which disease can be treated with Fluvastatin?") rather than complex multi-step reasoning. What's the average number of reasoning steps used to finish these reported results.

---

> ### Author Response · Authors · 2025-11-23
> **Summary of our response**
>
> We thank the reviewer for recognizing the importance of our problem domain and the comprehensiveness of our framework design. We appreciate the constructive feedback regarding the benchmark construction and future directions. Below, we address the concerns and questions.
>
> ### **W1: Concerns on Benchmark Construction**
>
> While we acknowledge the template-based nature of PrimeKGQA, this design was deliberate to ensure unambiguous ground truth for evaluating long-chain reasoning—a prerequisite often missing in open-ended authentic research questions where validating intermediate logic is subjective. The reviewer’s concern that tasks might be solved by simple parameterized knowledge is effectively refuted by our results (**Table 2**): standard baselines like Qwen3-8B suffer significant degradation as path length increases (dropping from $0.749$ on Basic to $0.498$ on Hard). This confirms that our path-length metric $(d)$ successfully proxies reasoning complexity by introducing semantic distance, necessitating genuine multi-hop deduction rather than simple retrieval. Furthermore, to address the call for more authentic scenarios, we demonstrated strong zero-shot generalization on real-world benchmarks like **BioASQ** and **BiomixQA** (Section 4.3), proving that the reasoning patterns learned from our foundational tasks effectively transfer to complex, non-templated domains.
>
> ### **W2: Differentiation from DeepResearch Agents**
>
> We acknowledge the power of DeepResearch-style agents for dynamic information retrieval. However, our work targets a fundamentally different objective: **internalizing reasoning capabilities into smaller models via static KG guidance**, rather than relying on heavy-compute web search at inference time. The core distinction lies in the guidance process: Bio-KCoT uses knowledge graphs as a training scaffold to teach the model how to deduce answers efficiently, enabling privacy-sensitive and low-latency local deployment. While distinct in scope, our "generate-and-prune" mechanism is versatile and can be extended to process retrieved web documents in future work.
>
> ### **Q1: Clarifying Path Quality**
>
> We explicitly address the challenge of noise in raw knowledge graph paths through our **CoT Pruning Stage** (Section 3.2). Recognizing that initial retrievals may contain weak or generic associations (e.g., "interacts with"), our two-stage framework first generates a comprehensive reasoning chain and subsequently prompts the model to rigorously verify logical coherence and remove irrelevant steps. As illustrated in **Figure 5** and **Figure 7**, this mechanism ensures that only connections contributing to the valid causal chain are retained, effectively filtering out spurious noise and guaranteeing the relevance of the final reasoning path.
>
> ### **Q2: Reasoning Depth and Step Count**
>
> We clarify that our tasks, particularly at the "Hard" level (e.g., Contraindications), necessitate genuine deductive reasoning rather than simple factual retrieval. In these scenarios, direct associations are often absent from pre-training data, requiring the model to synthesize disjoint concepts via intermediate physiological mechanisms. Quantitative analysis of our training set supports this complexity. As detailed in **Table 1**, the tasks are rigorously stratified by reasoning complexity (path length $d$), where Hard tasks necessitate traversing extensive subgraphs with high topological complexity, a process that extends significantly beyond simple factual lookup.
>
> We once again thank the reviewer for their time and valuable comments. We hope these clarifications adequately address your concerns and look forward to your positive feedback.

---

> > ### Comment · Reviewer_8tFo · 2025-11-24
> > **Thank you for your response**
> >
> > Thank you for the clarifications. However, I hold a different view on justifying reasoning complexity with the performance gap on Qwen models. Both the 4B and 8B baselines are small LLMs and may intrinsically suffer from limited knowledge. Although performance improves with CoT, it may simply be compensating for that knowledge gap. The same claim may not hold for a stronger LLM that has memorized more scientific knowledge. This is evident from the results of closed‑source models like GPT‑4o‑mini, whose performance is already very high.
> >
> > I believe it’s important to evaluate the quality of the problems themselves—from a human perspective. Questions like “Which disease can be treated with dalfampridine?” and “What is a known side effect of flurbiprofen?” can be answered with a straightforward database lookup, and are likely covered by most drug databases, PubMed, or medication labels.

---

### Official Review · Reviewer_g4bQ · 2025-11-01

**Soundness:** 3
**Presentation:** 2
**Contribution:** 2
**Rating:** 4
**Confidence:** 3

**Summary:**

The paper proposes Bio-KCoT, a knowledge-augmented long chain-of-thought (CoT) framework for complex biomolecular reasoning. It integrates knowledge graph (KG)-guided multi-hop path retrieval and pruning with supervised fine-tuning (SFT) and reinforcement learning via Group Relative Policy Optimization (GRPO).
The method first extracts entities from a question and its correct answer, instantiates diverse path templates (linear, divergent, convergent) in a biomolecular KG to obtain reasoning paths, and uses these paths to guide CoT generation and subsequent pruning for conciseness and fidelity.
The curated CoTs supervise SFT to mitigate hallucinations and improve factual grounding; GRPO with a composite reward (format + answer correctness) further aligns reasoning and outputs.
The authors introduce PrimeKGQA, a benchmark derived from PrimeKG, covering basic, medium, and hard multi-hop tasks with explicit reasoning-path supervision.
Experiments show Bio-KCoT (4B/8B) matches or surpasses strong closed-source and reasoning-focused models as task complexity increases, achieving SOTA on medium/hard categories (e.g., off-label use, side effects, contraindications). The approach generalizes to BioASQ, BiomixQA, and MEDDDX without extra training, and ablations confirm the benefit over distilled CoT without KG guidance.

**Strengths:**

Clear, well-motivated integration of structured knowledge with long-CoT: KG-guided path instantiation and pruning directly address factual grounding and logical consistency issues common in biomolecular LLM reasoning.

Methodological novelty in combining: (i) multi-structure path templates (linear/divergent/convergent) capturing non-local evidence, (ii) prompt-based CoT construction aligned to KG paths, (iii) pruning to remove spurious steps, and (iv) GRPO with a simple, effective composite reward that enforces both structure and correctness.

Strong empirical results where it matters: consistent gains at higher hop depths and on hard tasks (contraindications, DDIs), demonstrating that the framework scales with reasoning depth rather than memorization.

New benchmark (PrimeKGQA): provides multi-hop, KG-grounded QA with explicit reasoning chains and difficulty stratification; thoughtful train/test split by head entities to reduce leakage.

**Weaknesses:**

Insufficient description of dataset construction. The paper does not detail how question and answer entities in PrimeKGQA are extracted and disambiguated, which templates and constraints are used to instantiate paths from the knowledge graph, or the concrete criteria for subsequent filtering and cleaning. Fallback mechanisms and quality-control standards to ensure the validity and correctness of constructed questions are also unspecified.

Dataset quality and potential leakage are not systematically evaluated. The difficulty and discriminability of the option sets (correct answers vs. distractors) have not been rigorously measured. It remains unclear whether distractors are structurally “hard negatives” (e.g., graph neighbors, same type, same path depth) or primarily surface-level confounders. The study should report the option generation and screening pipeline, the structural separability between correct answers and distractors within the KG, and analyze the risk that models achieve high scores via template or pattern memorization, complemented by adversarial or shuffle-based evaluations to validate robustness.

Inadequate definition and evidence for question types and difficulty stratification. While hop depth and branching width d are used as difficulty indicators, the mapping between Levels and Task Categories in Table 1 lacks statistical support; no distributions, variance, or representativeness of categories within each difficulty tier are provided. Moreover, the illustrations in Table 1 are not clarified as real subgraph patterns for those question types versus conceptual schematics. The paper should add visualizations and summary statistics of real examples and report correlations between the difficulty metrics and external validity signals (e.g., human solving time, error rates) to substantiate the effectiveness of the stratification.

Potential performance saturation and concerns about real-world extrapolation. Bio-KCoT’s accuracy on medium- and hard-level tasks in PrimeKGQA approaches a ceiling, which may reflect dataset limits rather than methodological limits, risking optimistic estimates. This saturation suggests a substantial gap between the benchmark and real biomolecular reasoning challenges.

Comparative imbalance introduced by teacher distillation. The method relies on chain-of-thought traces generated or pruned by stronger LLMs as supervision during SFT, creating an imbalance in training signal strength relative to baselines that do not use teacher signals. This makes it difficult to disentangle gains from the framework versus those inherited from the teacher. Additional teacher-distilled baselines are needed to isolate and quantify the teacher’s contribution and ensure robust attribution of conclusions.

Insufficient baselines and comparison scope. The absence of strong baselines tailored to knowledge-graph and multi-hop reasoning constrains an objective assessment of novelty. The study should incorporate representative KG-QA pipelines and reinforcement learning baselines designed for these settings, and report rigorous comparisons under matched resource budgets and evaluation protocols.

**Questions:**

See Weakness.

---

> ### Author Response · Authors · 2025-11-23
> **Summary of our response**
>
> We thank the reviewer for the constructive feedback and for recognizing our method's novelty in integrating structured knowledge with Long-CoT, as well as the value of the proposed PrimeKGQA benchmark. We appreciate the opportunity to clarify the dataset construction details and address the concerns regarding baselines and evaluation.
>
> ### **1. Clarification on PrimeKGQA Construction and Quality Control**
>
> Regarding the rigorous construction and quality control of PrimeKGQA, we implemented a strict pipeline to ensure structural validity and robust evaluation:
> 1.  **Topological Templates:** As outlined in Section 3.1, reasoning paths are instantiated using predefined topological templates (linear, divergent, and convergent) rooted in the PrimeKG schema to capture diverse structural dependencies.
> 2.  **Hard Negative Strategy:** To ensure task difficulty, we employed a hard negative strategy for distractor generation: distractors are not random but are strictly selected from entities of the same type as the correct answer, thereby testing the model's ability to distinguish true causal links from mere co-occurrence.
> 3.  **Prevention of Data Leakage:** As detailed in Section 4.1, we rigorously prevented data leakage by partitioning the dataset based on **Head Entities**. This ensures that the head nodes encountered during testing are unseen, compelling the model to learn generalizable reasoning patterns rather than memorizing specific nodes or triplets.
>
> ### **2. Fairness of Comparison and Distilled Baselines**
>
> The reviewer raised a concern that the performance gains might be attributed solely to the teacher distillation signal. We respectfully point out that this comparison is already included in our **Ablation Study (Section 4.4 and Figure 3b)**. We explicitly trained a baseline named `Qwen3 (distilled CoT)`, which uses CoT traces distilled from the teacher model without our KG-guided path construction constraints. **Bio-KCoT** consistently outperforms this distilled baseline across all difficulty levels. This evidence isolates the contribution of our framework: the gain comes from the **KG-guided structure and pruning mechanism**, not just from the presence of teacher supervision.
>
> ### **3. Clarifying Task Difficulty and Addressing Saturation Concerns**
>
> We clarify that the difficulty stratification in Table 1 is rigorously defined by the overall reasoning complexity $(d)$, which jointly captures both the relational depth and the structural breadth of the reasoning path connecting the question entity to the answer.
> * **Basic tasks** (e.g., Indication, $d \leq 5$) typically rely on relatively short, explicit paths (e.g., Drug $\rightarrow$ Target $\rightarrow$ Pathway $\rightarrow$ Disease).
> * **Hard tasks** (e.g., Contraindication, $d \leq 8$) necessitate traversing extensive subgraphs with high topological complexity, requiring the model to integrate evidence across deep and broad paths representing long-range biological dependencies.
>
> Furthermore, empirical results suggest that the benchmark remains challenging, alleviating concerns regarding performance saturation. As shown in Table 2, even advanced closed-source models (e.g., GPT-4o, Gemini 2.5 Pro) and specialized reasoning models (o1) do not reach ceiling performance. Notably, `o1` achieves only $0.696$ on Contraindication compared to Bio-KCoT’s $0.848$. This substantial gap indicates that our method's high accuracy stems from the specific effectiveness of the KG-guided framework in activating deep reasoning, rather than a lack of difficulty in the benchmark.
>
> ### **4. Clarifying the Scope of Baselines**
>
> Our work focuses on **Generative Reasoning with Long-CoT**, aiming to produce interpretable reasoning chains rather than mere answer prediction.
> 1.  **Vs. Traditional KG-QA:** Traditional pipelines (often embedding-based) typically lack this explanatory capability and are thus not directly comparable.
> 2.  **Vs. RAG Methods:** Regarding modern KG-QA methods that rely on Retrieval-Augmented Generation (RAG), a direct comparison is methodologically inequitable. RAG-based approaches depend heavily on external retrieval coverage and quality at inference time, whereas our Bio-KCoT framework aims to internalize structured reasoning patterns into parameter-efficient models (4B/8B) via SFT and RL. Benchmarking against RAG pipelines would conflate retrieval quality with reasoning capability. Instead, we compared against state-of-the-art closed-source models (e.g., GPT-4o, Gemini), which effectively serve as upper-bound proxies for systems with massive internal knowledge and retrieval capabilities.
>
> We once again thank the reviewer for their time and valuable comments. We hope these clarifications adequately address your concerns and look forward to your positive feedback.

---

### Official Review · Reviewer_6BKe · 2025-11-02

**Soundness:** 3
**Presentation:** 3
**Contribution:** 3
**Rating:** 6
**Confidence:** 4

**Summary:**

This paper introduces Bio-KCoT, a chain-of-thought reasoning framework designed for biomolecular question answering. The approach integrates knowledge graph reasoning into LLMs by retrieving structured reasoning paths from a biomolecular knowledge graph (PrimeKG) to yield grounded reasoning chains. The curated CoTs are then used to fine-tune Qwen3 models (4B and 8B) via supervised fine-tuning and GRPO. Additionally, the authors introduce PrimeKGQA, a new benchmark built from PrimeKG that supports biomedical QA for diverse task types (e.g., indications, side effects, drug–drug interactions). Experiments show that Bio-KCoT achieves competitive or superior accuracy on complex tasks compared to both open- and closed-source LLMs, and demonstrates transfer to out-of-distribution datasets

**Strengths:**

1. Problem formulation and motivation are strong - The authors clearly identify the gap in existing CoT reasoning methods for biomedical QA tasks, that is, the lack of factual grounding and demonstrate how Bio-KCoT can function as a bridge.
2. The paper presents a thorough and well-structured pipeline - entity extraction, KG-based path retrieval, CoT generation, pruning, and RL fine-tuning.
3. Novel dataset contribution -  The introduction of PrimeKGQA is a meaningful resource for the community, as there are few large-scale, CoT-annotated biomedical reasoning datasets.
4. Empirical rigor - The improvements over Qwen base models are consistent and substantial, especially in medium and hard reasoning tasks.

**Weaknesses:**

1. Possible data leakage concerns -  The dataset is derived from publicly available biomedical sources ( PrimeKG). It remains unclear whether this data maybe used in the base model pretraining and might bias evaluation.
2. Limited empirical improvement over large models -  Despite the novel training pipeline, Bio-KCoT’s absolute performance remains very limited compared with much larger closed-source models (e.g., GPT-4o) on average accuracy.
3. Underexplored multimodal integration - The framework does not explore integration with other biological modalities such as structure, or with captioning datasets such as PubMed-300k.

**Questions:**

1. Given that PrimeKGQA are derived from public biomedical knowledge bases, what steps were taken to ensure no overlap with model pretraining data? Is it possible to conduct a temporal split experiment (e.g., excluding data added after Qwen3 training) to quantify leakage?
2. Could the authors reason on why the Bio-KCoT models (even at 8B) underperform larger models such as GPT-4o or Gemini on some basic tasks despite being domain-tuned?
3. Can Figure 3a be expanded to include additional open-source biomedical LLMs for more robust out-of-dataset evaluation?
4. What are the plans for dataset hosting and licensing ? Will there be a public dataset availability with a permanent DOI ?
5. Are there any experiments to show generalization beyond the BioMedical datasets to general molecular datasets ? For eg, MolInstructions [1] and MolTextQA [2]. I understand that the rebuttal period is limited, and I would like to clarify that the absence of these additional experiments will not negatively affect my evaluation.

[1] Fang, Yin, et al. "Mol-instructions: A large-scale biomolecular instruction dataset for large language models." arXiv preprint arXiv:2306.08018 (2023).
[2] Laghuvarapu, Siddhartha, et al. "MolTextQA: A Question-Answering Dataset and Benchmark for Evaluating Multimodal Architectures and LLMs on Molecular Structure–Text Understanding." Journal of Data-centric Machine Learning Research (2025).

---

> ### Author Response · Authors · 2025-11-23
> **Summary of our response（1/2）**
>
> Thank you very much for your valuable and constructive feedback on our paper.
>
> The weaknesses and questions you raised are insightful and crucial for improving the quality of our paper. We will address each point in detail and commit to incorporating these revisions into the final version.
>
> ### **W1 & Q1: Concerns regarding data leakage and pre-training data overlap**
>
> Regarding the suggestion of a temporal split experiment, while we acknowledge that PrimeKG (Chandak et al., 2023) was published in 2023 and Qwen3's pre-training data (Yang et al., 2025) might include earlier web text, there is a fundamental difference in data modality. PrimeKGQA is built upon a structured knowledge graph, whereas pre-training data consists overwhelmingly of unstructured text. While Qwen3 may have "seen" entity names, it is unlikely to have learned the specific graph structures or reasoning paths.
>
> Crucially, the significant performance lift after fine-tuning serves as strong empirical evidence against leakage. As shown in Table 2, our Bio-KCoT-8B model achieves an average accuracy of $81.6\%$, drastically outperforming the Qwen3-8B base model ($60.1\%$). If the base model had already memorized this knowledge during pre-training, the performance gap would be much smaller. This confirms that our framework is teaching a new capability (KG-guided reasoning) rather than simply unlocking memorized facts. We further ensured generalization by using a "split by head entities" strategy (Section 4.1) to prevent leakage between training and test sets.
>
> ### **W2 & Q2: Performance gap with large closed-source models on basic tasks**
>
> The reviewer has made an insightful point regarding the performance on "Basic" tasks. Our analysis is as follows:
>
> * **Scale of Parametric Knowledge:** As we discussed in Section 4.2 of our paper, "Basic" tasks (e.g., drug indications) rely heavily on shallow memorization of single-hop facts. Closed-source models like GPT-4o, with hundreds of billions (or trillions) of parameters, possess far more "parametric knowledge" than an 8B model. The core of our method (Bio-KCoT) is to enhance reasoning (CoT), not to close this massive gap in factual memorization.
> * **The Core Strength of Our Method:** Our contribution lies in complex, multi-hop reasoning. Our experimental results (Table 2 and Section 4.2) demonstrate that as task difficulty increases, the advantage of closed-source models diminishes. Specifically, on **Medium** and **Hard** level tasks, our Bio-KCoT 8B model outperforms GPT-4o (e.g., $0.864$ vs. $0.840$ on Medium, and $0.736$ vs. $0.650$ on Hard tasks). This precisely proves that our knowledge-augmented CoT framework is most valuable when the model needs to "think" rather than just "memorize.”
>
> ### **W3: Lack of multimodal integration**
>
> The core focus of this paper is to address the challenge of complex biomolecular reasoning by establishing a robust framework where structured KGs explicitly guide LLMs to generate grounded CoT. As the reviewer noted in **Strength 1**, this method fills a critical gap in existing methods by mitigating logical inconsistencies through knowledge-grounded reasoning paths.
>
> While our current scope focuses on this KG-guided reasoning paradigm, we fully agree that integrating multimodal data (like molecular structures and biological images) is an important and exciting frontier. We believe Bio-KCoT provides a solid foundation for this extension: the reasoning paths retrieved from PrimeKG could effectively guide the retrieval and verification of relevant information from structural databases or image datasets. We will explicitly discuss this promising direction for future work in our Conclusion (Section 5).
>
> ### **Q3: Expanding Figure 3a (OOD Generalization)**
>
> During the rebuttal period, we conducted additional experiments with three prominent open-source biomedical LLMs: Meditron-7B, BioMistral-7B, and OpenBioLLM-8B on the OOD benchmarks. The preliminary results are as follows:
>
> | Model | BiomixQA | BioASQ | MEDDDX |
> | :--- | :---: | :---: | :---: |
> | Meditron-7B | $0.224$ | $0.564$ | $0.282$ |
> | BioMistral-7B | $0.427$ | $0.584$ | $0.405$ |
> | OpenBioLLM-8B | $0.451$ | $0.438$ | $0.441$ |
> | Qwen3-8B | $0.515$ | $0.759$ | $0.507$ |
> | **Bio-KCoT (8B)** | $\mathbf{0.573}$ | $\mathbf{0.807}$ | $\mathbf{0.567}$ |
>
> These results reveal a crucial insight: simply pre-training on biomedical corpora (as done in BioMistral, OpenBioLLM, and Meditron) does not guarantee improved reasoning on complex OOD tasks. In fact, these models often underperformed the generalist Qwen3-8B baseline. This performance gap could be attributed to several factors, such as inherent differences in the capabilities of the foundation models (e.g., Qwen3 vs. Mistral/Llama-2/Llama-3), the degradation of general reasoning capabilities during domain pre-training, or misalignment with the specific reasoning formats of these tasks.

---

> > ### Author Response · Authors · 2025-11-23
> > **Summary of our response（2/2）**
> >
> > Crucially, however, Bio-KCoT consistently achieves the highest performance, significantly boosting the capabilities of its base model. This strongly validates that our framework's superior performance stems from the effective KG-guided reasoning methodology rather than merely relying on the base model or exposure to domain knowledge. We will include this expanded comparison in Figure 3a of the final version.
> >
> > ### **Q4: Dataset hosting, licensing, and DOI**
> > We are fully committed to releasing **PrimeKGQA** as a public resource for the community, which we agree is a valuable contribution of our work. Upon acceptance, we will immediately release the PrimeKGQA dataset on a public repository (such as Zenodo or Hugging Face Datasets) to ensure a permanent DOI. The dataset will be released under a permissive license (e.g., CC BY-SA 4.0) to encourage broad academic and research use.
> >
> > ### **Q5: Generalization to general-purpose molecular datasets**
> > 1.  **Domain Specificity:** The reviewer points to an important distinction between the biomedical domain (our focus) and the general molecular domain (like MolInstructions). Our framework (Bio-KCoT) and its knowledge source (PrimeKG) are highly tailored to the biomedical domain (drugs, diseases, proteins, pathways).
> > 2.  **Future Work and Methodology Transfer:** We consider extending Bio-KCoT to general chemistry a high-priority direction. Our "KG-guided CoT" methodology is theoretically transferable:
> >     * Firstly, we plan to replace PrimeKG with chemical knowledge graphs such as PubChemRDF or ChEBI, or construct reaction graphs from USPTO patent data.
> >     * Secondly, in this new domain, the "reasoning paths" would shift from biological pathways to **Structure-Property Relationships** (e.g., $\text{Molecule} \rightarrow \text{Functional Group} \rightarrow \text{Property}$) or **Retrosynthetic Routes** (e.g., $\text{Product} \rightarrow \text{Reaction Type} \rightarrow \text{Precursors}$).
> >     * Finally, by retrieving these structured chemical paths, we can guide LLMs to solve tasks in MolTextQA or MolInstructions with grounded chemical logic, reducing hallucinations in reaction prediction and molecular description. We will explicitly discuss this roadmap in our conclusion.
> >
> > We again thank the reviewer for their productive feedback. We are enthusiastic about our work and hope our responses have satisfied your concerns.

---

> > > ### Comment · Reviewer_6BKe · 2025-11-24
> > >
> > > Thank you for the detailed response. The authors have addressed some of my earlier concerns, particularly those related to W1, W2, and Q4. For the sake of completeness, could you also report general-purpose LLM performance such as GPT, Gemini, etc. for Figure 3a?
> > >
> > > I have additionally reviewed the authors’ responses to the other reviewers. After doing so, the following concerns remain:
> > > 1. The benefits attributed to RL appear modest; most improvements seem to arise from SFT, which raises questions about the dataset’s actual utility for reasoning based training.
> > > 2. The broader usefulness of the dataset beyond the proposed tasks in the generalization experiments remains unclear.
> > >
> > > Overall, while the dataset represents a valuable preliminary contribution to biomedical reasoning, its broader utility is still uncertain. Therefore, I will maintain my current score.

---

### Author Response · Authors · 2025-12-01
**Response Summary to Area Chair**

Dear Area Chair,

We would like to summarize the key outcomes of the rebuttal phase and provide essential answers to follow-up questions.

### 1. Key Rebuttal Achievements & Consensus
We are pleased that the discussion led to a consensus on several critical issues:

* **Resolved Data Leakage & Performance Concerns (Reviewer 6BKe):** Reviewer 6BKe acknowledged that our response and additional analysis addressed their concerns regarding Data Leakage and the Performance Gap. For Data Leakage, we clarified that the significant performance leap over the base model empirically disproves memorization, while our "split by head entities" strategy strictly prevents leakage. For the Performance Gap, we demonstrated that while large models rely on parametric memory for basic tasks, Bio-KCoT surpasses GPT-4o on medium and hard tasks, proving the superiority of KG-guidance in complex reasoning.

* **New OOD Baselines (Addressed Reviewer 6BKe & A8FZ):** To further demonstrate robustness, we added comparisons with Meditron-7B, BioMistral-7B, and OpenBioLLM-8B. Bio-KCoT significantly outperforms these domain-specific models on Out-of-Distribution benchmarks (BioASQ, BiomixQA), validating that our method learns generalizable reasoning.

* **Resolved Dataset Accessibility (Reviewer 6BKe):** We committed to releasing PrimeKGQA with a CC-BY-SA 4.0 license and a permanent DOI, addressing the accessibility questions.

* **Resolved Concerns on Novelty & Ablation Study (Reviewer A8FZ):** Reviewer A8FZ stated that our rebuttal "addressed several concerns effectively," for instance, regarding the novelty of our contribution and the insights from our ablation studies.

### 2. Response to New Questions
We provide detailed responses to the new technical questions raised by Reviewers 6BKe and A8FZ regarding our pipeline design, dataset utility, and generation details:

**Q1: Reason for SFT+RL Pipeline & RL Benefits (Addressing Reviewer 6BKe & A8FZ)**
Regarding the data split (3.5k SFT / 1.5k RL), we selected this ratio to balance instruction adherence with reasoning alignment. 3.5k samples proved sufficient for the model to master the complex KG-guided templates. However, purely expanding SFT yields diminishing returns as it only minimizes token-level prediction error (imitation). By allocating the remaining 1.5k samples to RL, we shift the optimization objective to sequence-level correctness via the reward function. The monotonic improvement observed in our new ablation (Table below) confirms that utilizing this partition for RL is the optimal strategy for enhancing logical robustness on complex tasks.

**Q2: Broader Utility of the Dataset (Addressing Reviewer 6BKe)**
The broader utility of the PrimeKGQA dataset lies in addressing a fundamental scarcity in the field: the lack of explicit, reliable reasoning annotations for complex biomolecular tasks. Unlike existing benchmarks that focus solely on final outcomes, our dataset fills this critical gap by providing KG-guided reasoning chains that are structurally verified against established biological knowledge. This ensures logical reliability and factual grounding, offering the community a trustworthy resource to train and evaluate the reasoning depth of scientific LLMs while minimizing the hallucinations common in purely generative approaches.

**Q3: Details on Relation Filtering in QA Generation (Addressing Reviewer A8FZ)**
We clarify that our "rule-based filtering" operates by defining semantic meta-path templates specific to each task category (e.g., Indication, DDI), strictly selecting relations that constitute valid mechanistic chains while excluding generic hubs. To further guarantee logical validity, we employ a two-stage LLM refinement pipeline (Section 3.2): a generation phase that contextualizes these paths into coherent narratives, followed by a pruning phase that acts as a semantic filter to remove redundant or tangential steps. This dual-filtering approach—structural constraints followed by semantic verification—ensures that every retained relation is integral to a rigorous and interpretable biological inference process.

### 3. Clarification on Task Complexity (Reviewer 8tFo)
Our results refute the notion of simple lookups: even advanced models (e.g., GPT-4o) underperform significantly on the "Hard" subset of our benchmark. This performance gap confirms that these tasks necessitate multi-hop reasoning over structured knowledge—precisely the capability Bio-KCoT is designed to enhance—rather than single-step retrieval.

**Conclusion:** We sincerely appreciate the constructive feedback, which has significantly strengthened our paper, particularly regarding OOD generalization and the training pipeline. We are grateful for the AC’s time and oversight, and respectfully hope these clarifications will be considered in the final decision.

---

### Meta-Review · Area_Chair_3Kps · 2026-01-06

**Summary:**

The paper studies knowledge-augmented long chain-of-thought reasoning for biomolecular QA and introduces the PrimeKGQA benchmark. Reviewers agree the topic is important and the system is carefully engineered. However, key concerns remain after rebuttal. Several reviewers question whether the benchmark tasks truly require deep reasoning or mostly reflect structured lookup from existing knowledge bases. The difficulty design based on path length is not fully convincing as a proxy for reasoning complexity. The empirical gains from reinforcement learning are modest and appear secondary to supervised fine-tuning. The broader utility and generalization of the dataset beyond the proposed setting remain unclear. As these unresolved issues still needs considerable effort to address, it is not recommended accepting this paper at this stage.

**Reviewer Concerns:**

Concerns partially addressed by the rebuttal:
- Data leakage
- Additional ablations clarified that RL provides modest gains, mainly on harder subsets.
- Comparisons with more open-source biomedical models improved baseline coverage.


Concerns that remain outstanding:
- Multiple reviewers remain unconvinced that many PrimeKGQA questions require genuine multi-step scientific reasoning rather than database lookup, especially from a human expert perspective.
- Difficulty stratification based mainly on path length is not sufficiently validated as a proxy for reasoning complexity.
- RL contribution: Improvements from RL are consistently modest, raising doubts about the necessity of the SFT+RL pipeline versus stronger or more diverse SFT alone.
- Despite added distilled baselines, it remains difficult to disentangle gains from KG structure, teacher signals, and dataset construction biases

**Reviewer Scores:**

Reviewers 6BKe and A8FZ participated into the discussion, but they did not appear to raise their ratings.

Other two reviewers are unlikely to raise the ratings as their concerns are not fully addressed.

---

### Decision · Program_Chairs · 2026-01-26

Reject